# ResNEsts and DenseNEsts: Block-based DNN Models with Improved Representation Guarantees

**Kuan-Lin Chen[1], Ching-Hua Lee[1], Harinath Garudadri[2], and Bhaskar D. Rao[1]**

[1]Department of Electrical and Computer Engineering, [2]Qualcomm Institute
University of California, San Diego
La Jolla, CA 92093, USA
{kuc029,chl438,hgarudadri,brao}@ucsd.edu

## Abstract

Models recently used in the literature proving residual networks (ResNets) are better than linear predictors are actually different from standard ResNets that have been widely used in computer vision. In addition to the assumptions such as scalar-valued output or single residual block, the models fundamentally considered in the literature have no nonlinearities at the final residual representation that feeds into the final affine layer. To codify such a difference in nonlinearities and reveal a linear estimation property, we define ResNEsts, i.e., Residual Nonlinear Estimators, by simply dropping nonlinearities at the last residual representation from standard ResNets. We show that wide ResNEsts with bottleneck blocks can always guarantee a very desirable training property that standard ResNets aim to achieve, i.e., adding more blocks does not decrease performance given the same set of basis elements. To prove that, we first recognize ResNEsts are basis function models that are limited by a coupling problem in basis learning and linear prediction. Then, to decouple prediction weights from basis learning, we construct a special architecture termed augmented ResNEst (A-ResNEst) that always guarantees no worse performance with the addition of a block. As a result, such an A-ResNEst establishes empirical risk lower bounds for a ResNEst using corresponding bases. Our results demonstrate ResNEsts indeed have a problem of diminishing feature reuse; however, it can be avoided by sufficiently expanding or widening the input space, leading to the above-mentioned desirable property. Inspired by the densely connected networks (DenseNets) that have been shown to outperform ResNets, we also propose a corresponding new model called Densely connected Nonlinear Estimator (DenseNEst). We show that any DenseNEst can be represented as a wide ResNEst with bottleneck blocks. Unlike ResNEsts, DenseNEsts exhibit the desirable property without any special architectural re-design.

## 1 Introduction

Constructing deep neural network (DNN) models by stacking layers unlocks the field of deep learning, leading to the early success in computer vision, such as AlexNet [Krizhevsky et al., 2012], ZFNet [Zeiler and Fergus, 2014], and VGG [Simonyan and Zisserman, 2015]. However, stacking more and more layers can suffer from worse performance [He and Sun, 2015, Srivastava et al., 2015, He et al., 2016a]; thus, it is no longer a valid option to further improve DNN models. In fact, such a *degradation problem* is not caused by overfitting, but worse training performance [He et al., 2016a]. When neural networks become sufficiently deep, optimization landscapes quickly transition from being nearly convex to being highly chaotic [Li et al., 2018]. As a result, stacking more and more layers in DNN models can easily converge to poor local minima (see Figure 1 in [He et al., 2016a]).

35th Conference on Neural Information Processing Systems (NeurIPS 2021).

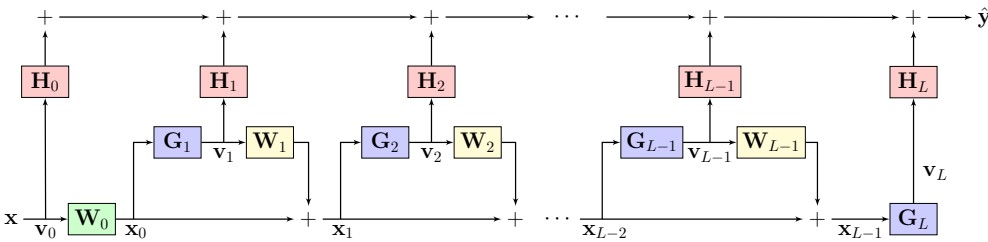

Figure 1: The proposed augmented ResNEst or A-ResNEst. A set of new prediction weights $\mathbf{H}_0, \mathbf{H}_1, \cdots, \mathbf{H}_L$ are introduced on top of the features in the ResNEst (see Figure 2). The A-ResNEst is always better than the ResNEst in terms of empirical risk minimization (see Proposition 2). Empirical results of the A-ResNEst model are deferred to Appendix B in the supplementary material.

To address the issue above, the modern deep learning paradigm has shifted to designing DNN models based on blocks or modules of the same kind in cascade. A block or module comprises specific operations on a stack of layers to avoid the degradation problem and learn better representations. For example, Inception modules in the GoogLeNet [Szegedy et al., 2015], residual blocks in the ResNet [He et al., 2016a,b, Zagoruyko and Komodakis, 2016, Kim et al., 2016, Xie et al., 2017, Xiong et al., 2018], dense blocks in the DenseNet [Huang et al., 2017], attention modules in the Transformer [Vaswani et al., 2017], Squeeze-and-Excitation (SE) blocks in the SE network (SENet) [Hu et al., 2018], and residual U-blocks [Qin et al., 2020] in $U^2$-Net. Among the above examples, the most popular block design is the residual block which merely adds a skip connection (or a residual connection) between the input and output of a stack of layers. This modification has led to a huge success in deep learning. Many modern DNN models in different applications also adopt residual blocks in their architectures, e.g., V-Net in medical image segmentation [Milletari et al., 2016], Transformer in machine translation [Vaswani et al., 2017], and residual LSTM in speech recognition [Kim et al., 2017]. Empirical results have shown that ResNets can be even scaled up to 1001 layers or 333 bottleneck residual blocks, and still improve performance [He et al., 2016b].

Despite the huge success, our understanding of ResNets is very limited. To the best of our knowledge, no theoretical results have addressed the following question: *Is learning better ResNets as easy as stacking more blocks?* The most recognized intuitive answer for the above question is that a particular stack of layers can focus on fitting the residual between the target and the representation generated in the previous residual block; thus, adding more blocks always leads to no worse training performance. Such an intuition is indeed true for a constructively blockwise training procedure; but not clear when the weights in a ResNet are optimized as a whole. Perhaps the theoretical works in the literature closest to the above question are recent results in an albeit modified and constrained ResNet model that every local minimum is less than or equal to the empirical risk provided by the best linear predictor [Shamir, 2018, Kawaguchi and Bengio, 2019, Yun et al., 2019]. Although the aims of these works are different from our question, they actually prove a special case under these simplified models in which the final residual representation is better than the input representation for linear prediction. We notice that the models considered in these works are very different from standard ResNets using pre-activation residual blocks [He et al., 2016b] due to the absence of the nonlinearities at the final residual representation that feeds into the final affine layer. Other noticeable simplifications include scalar-valued output [Shamir, 2018, Yun et al., 2019] and single residual block [Shamir, 2018, Kawaguchi and Bengio, 2019]. In particular, Yun et al. [2019] additionally showed that residual representations do not necessarily improve monotonically over subsequent blocks, which highlights a fundamental difficulty in analyzing their simplified ResNet models.

In this paper, we take a step towards answering the above-mentioned question by constructing practical and analyzable block-based DNN models. Main contributions of our paper are as follows:

**Improved representation guarantees for wide ResNEsts with bottleneck residual blocks.** We define a ResNEst as a standard single-stage ResNet that simply drops the nonlinearities at the last residual representation (see Figure 2). We prove that sufficiently wide ResNEsts with bottleneck residual blocks under practical assumptions can always guarantee a desirable training property that ResNets with bottleneck residual blocks empirically achieve (but theoretically difficult to prove), i.e., adding more blocks does not decrease performance given the same arbitrarily selected basis.

To be more specific, any local minimum obtained from ResNEsts has an improved representation guarantee under practical assumptions (see Remark 2 (a) and Corollary 1). Our results apply to loss functions that are differentiable and convex; and do not rely on any assumptions regarding datasets, or convexity/differentiability of the residual functions.

**Basic vs. bottleneck.** In the original ResNet paper, He et al. [2016a] empirically pointed out that ResNets with basic residual blocks indeed gain accuracy from increased depth, but are not as economical as the ResNets with bottleneck residual blocks (see Figure 1 in [Zagoruyko and Komodakis, 2016] for different block types). Our Theorem 1 supports such empirical findings.

**Generalized and analyzable DNN models.** ResNEsts are more general than the models considered in [Hardt and Ma, 2017, Shamir, 2018, Kawaguchi and Bengio, 2019, Yun et al., 2019] due to the removal of their simplified ResNet settings. In addition, the ResNEst modifies the input by *an expansion layer* that expands the input space. Such an expansion turns out to be crucial in deriving theoretical guarantees for improved residual representations. We find that the importance on expanding the input space in standard ResNets with bottleneck residual blocks has not been well recognized in existing theoretical results in the literature.

**Restricted basis function models.** We reveal a linear relationship between the output of the ResNEst and the input feature as well as the feature vector going into the last affine layer in each of residual functions. By treating each of feature vectors as a basis element, we find that ResNEsts are basis function models handicapped by *a coupling problem* in basis learning and linear prediction that can limit performance.

**Augmented ResNEsts.** As shown in Figure 1, we present a special architecture called *augmented ResNEst* or *A-ResNEst* that introduces a new weight matrix on each of feature vectors to solve the coupling problem that exists in ResNEsts. Due to such a decoupling, every local minimum obtained from an A-ResNEst bounds the empirical risk of the associated ResNEst from below. A-ResNEsts also directly enable us to see how features are supposed to be learned. It is necessary for features to be *linearly unpredictable* if residual representations are strictly improved over blocks.

**Wide ResNEsts with bottleneck residual blocks do not suffer from saddle points.** At every saddle point obtained from a ResNEst, we show that there exists at least one direction with strictly negative curvature, under the same assumptions used in the improved representation guarantee, along with the specification of a squared loss and suitable assumptions on the last feature and dataset.

**Improved representation guarantees for DenseNEsts.** Although DenseNets [Huang et al., 2017] have shown better empirical performance than ResNets, we are not aware of any theoretical support for DenseNets. We define a DenseNEst (see Figure 4) as a simplified DenseNet model that only utilizes the dense connectivity of the DenseNet model, i.e., direct connections from every stack of layers to all subsequent stacks of layers. We show that any DenseNEst can be represented as a wide ResNEst with bottleneck residual blocks equipped with orthogonalities. Unlike ResNEsts, any DenseNEst exhibits the desirable property, i.e., adding more dense blocks does not decrease performance, without any special architectural re-design. Compared to A-ResNEsts, the way the features are generated in DenseNEsts makes linear predictability even more unlikely, suggesting better feature construction.

## 2   ResNEsts and augmented ResNEsts

In this section, we describe the proposed DNN models. These models and their new insights are preliminaries to our main results in Section 3. Section 2.1 recognizes the importance of the expansion layer and defines the ResNEst model. Section 2.2 points out the basis function modeling interpretation and the coupling problem in ResNEsts, and shows that the optimization on the set of prediction weights is non-convex. Section 2.3 proposes the A-ResNEst to avoid the coupling problem and shows that the minimum empirical risk obtained from a ResNEst is bounded from below by the corresponding A-ResNEst. Section 2.4 shows that linearly unpredictable features are necessary for strictly improved residual representations in A-ResNEsts.

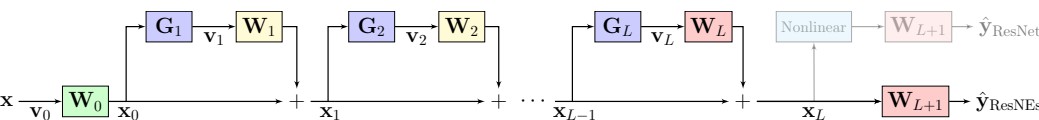

Figure 2: A generic vector-valued ResNEst that has a chain of $L$ residual blocks (or units). Redrawing the standard ResNet block diagram in this viewpoint gives us considerable new insight. The symbol "+" represents the addition operation. Different from the ResNet architecture using pre-activation residual blocks in the literature [He et al., 2016b], our ResNEst architecture drops nonlinearities at $\mathbf{x}_L$ so as to reveal a linear relationship between the output $\hat{\mathbf{y}}_{\text{ResNEst}}$ and the features $\mathbf{v}_0, \mathbf{v}_1, \cdots, \mathbf{v}_L$. Empirical results of the ResNEst model are deferred to Appendix B in the supplementary material.

## 2.1 Dropping nonlinearities in the final representation and expanding the input space

The importance on expanding the input space via $\mathbf{W}_0$ (see Figure 2) in standard ResNets has not been well recognized in recent theoretical results ([Shamir, 2018, Kawaguchi and Bengio, 2019, Yun et al., 2019]) although standard ResNets always have an expansion implemented by the first layer before the first residual block. Empirical results have even shown that a standard 16-layer wide ResNet outperforms a standard 1001-layer ResNet [Zagoruyko and Komodakis, 2016], which implies the importance of *a wide expansion of the input space*.

We consider the proposed ResNEst model shown in Figure 2 whose $i$-th residual block employs the following input-output relationship:

$$\mathbf{x}_i = \mathbf{x}_{i-1} + \mathbf{W}_i \mathbf{G}_i (\mathbf{x}_{i-1}; \boldsymbol{\theta}_i) \tag{1}$$

for $i = 1, 2, \cdots, L$. The term excluded the first term $\mathbf{x}_{i-1}$ on the right-hand side is a composition of a nonlinear function $\mathbf{G}_i$ and a linear transformation,[1] which is generally known as a residual function. $\mathbf{W}_i \in \mathbb{R}^{M \times K_i}$ forms a linear transformation and we consider $\mathbf{G}_i (\mathbf{x}_{i-1}; \boldsymbol{\theta}_i) : \mathbb{R}^M \mapsto \mathbb{R}^{K_i}$ as a function implemented by a neural network with parameters $\boldsymbol{\theta}_i$ for all $i \in \{1, 2, \cdots, L\}$. We define the expansion $\mathbf{x}_0 = \mathbf{W}_0 \mathbf{x}$ for the input $\mathbf{x} \in \mathbb{R}^{N_{in}}$ to the ResNEst using a linear transformation with a weight matrix $\mathbf{W}_0 \in \mathbb{R}^{M \times K_0}$. The output $\hat{\mathbf{y}}_{\text{ResNEst}} \in \mathbb{R}^{N_o}$ (or $\hat{\mathbf{y}}_{L\text{-ResNEst}}$ to indicate $L$ blocks) of the ResNEst is defined as $\hat{\mathbf{y}}_{L\text{-ResNEst}} (\mathbf{x}) = \mathbf{W}_{L+1} \mathbf{x}_L$ where $\mathbf{W}_{L+1} \in \mathbb{R}^{N_o \times M}$. $M$ is the expansion factor and $N_o$ is the output dimension of the network. The number of blocks $L$ is a nonnegative integer. When $L = 0$, the ResNEst is a two-layer linear network $\hat{\mathbf{y}}_{0\text{-ResNEst}} (\mathbf{x}) = \mathbf{W}_1 \mathbf{W}_0 \mathbf{x}$.

Notice that the ResNEst we consider in this paper (Figure 2) is more general than the models in [Hardt and Ma, 2017, Shamir, 2018, Kawaguchi and Bengio, 2019, Yun et al., 2019] because our residual space $\mathbb{R}^M$ (the space where the addition is performed at the end of each residual block) is not constrained by the input dimension due to the expansion we define. Intuitively, a wider expansion (larger $M$) is required for a ResNEst that has more residual blocks. This is because the information collected in the residual representation grows after each block, and the fixed dimension $M$ of the residual representation must be sufficiently large to avoid any loss of information. It turns out a wider expansion in a ResNEst is crucial in deriving performance guarantees because it assures the quality of local minima and saddle points (see Theorem 1 and 2).

## 2.2 Basis function modeling and the coupling problem

The conventional input-output relationship of a standard ResNet is not often easy to interpret. We find that redrawing the standard ResNet block diagram [He et al., 2016a,b] with a different viewpoint, shown in Figure 2, can give us considerable new insight. As shown in Figure 2, the ResNEst now reveals a linear relationship between the output and the features. With this observation, we can write down a useful input-output relationship for the ResNEst:

$$\hat{\mathbf{y}}_{L\text{-ResNEst}} (\mathbf{x}) = \mathbf{W}_{L+1} \sum_{i=0}^{L} \mathbf{W}_i \mathbf{v}_i (\mathbf{x}) \tag{2}$$

---

[1] For any affine function $\mathbf{y}(\mathbf{x}_{\text{raw}}) = \mathbf{A}_{\text{raw}} \mathbf{x}_{\text{raw}} + \mathbf{b}$, if desired, one can use $\mathbf{y}(\mathbf{x}) = \begin{bmatrix} \mathbf{A}_{\text{raw}} & \mathbf{b} \end{bmatrix} \begin{bmatrix} \mathbf{x}_{\text{raw}} \\ 1 \end{bmatrix} = \mathbf{A} \mathbf{x}$

where $\mathbf{A} = \begin{bmatrix} \mathbf{A}_{\text{raw}} & \mathbf{b} \end{bmatrix}$ and $\mathbf{x} = \begin{bmatrix} \mathbf{x}_{\text{raw}} \\ 1 \end{bmatrix}$ and discuss on the linear function instead. All the results derived in this paper hold true regardless of the existence of bias parameters.

where $\mathbf{v}_i(\mathbf{x}) = \mathbf{G}_i(\mathbf{x}_{i-1}; \boldsymbol{\theta}_i) = \mathbf{G}_i\left(\sum_{j=0}^{i-1} \mathbf{W}_j \mathbf{v}_j; \boldsymbol{\theta}_i\right)$ for $i = 1, 2, \cdots, L$. Note that we do not impose any requirements for each $\mathbf{G}_i$ other than assuming that it is implemented by a neural network with a set of parameters $\boldsymbol{\theta}_i$. We define $\mathbf{v}_0 = \mathbf{v}_0(\mathbf{x}) = \mathbf{x}$ as the linear feature and regard $\mathbf{v}_1, \mathbf{v}_2, \cdots, \mathbf{v}_L$ as nonlinear features of the input $\mathbf{x}$, since $\mathbf{G}_i$ is in general nonlinear. The benefit of our formulation (2) is that the output of a ResNEst $\hat{\mathbf{y}}_{L\text{-ResNEst}}$ now can be viewed as a linear function of all these features. Our point of view of ResNEsts in (2) may be useful to explain the finding that ResNets are ensembles of relatively shallow networks [Veit et al., 2016].

As opposed to traditional nonlinear methods such as basis function modeling (chapter 3 in the book by Bishop, 2006) where a linear function is often trained on a set of handcrafted features, the ResNEst jointly finds features and a linear predictor function by solving the empirical risk minimization (ERM) problem denoted as (P) on $(\mathbf{W}_0, \cdots, \mathbf{W}_{L+1}, \boldsymbol{\theta}_1, \cdots, \boldsymbol{\theta}_L)$. We denote $\mathcal{R}$ as the empirical risk (will be used later on). Indeed, one can view training a ResNEst as *a basis function modeling with a trainable (data-driven) basis* by treating each of features as a basis vector (it is reasonable to assume all features are not linearly predictable, see Section 2.4). However, unlike a basis function modeling, the linear predictor function in the ResNEst is not entirely independent of the basis generation process. We call such a phenomenon as *a coupling problem* which can handicap the performance of ResNEsts. To see this, note that feature (basis) vectors $\mathbf{v}_{i+1}, \cdots, \mathbf{v}_L$ can be different if $\mathbf{W}_i$ is changed (the product $\mathbf{W}_{L+1} \mathbf{W}_i \mathbf{v}_i$ is the linear predictor function for the feature $\mathbf{v}_i$). Therefore, the set of parameters $\boldsymbol{\phi} = \{\mathbf{W}_{i-1}, \boldsymbol{\theta}_i\}_{i=1}^L$ needs to be fixed to sufficiently guarantee that the basis is not changed with different linear predictor functions. It follows that $\mathbf{W}_{L+1}$ and $\mathbf{W}_L$ are the only weights which can be adjusted without changing the features. We refer to $\mathbf{W}_L$ and $\mathbf{W}_{L+1}$ as prediction weights and $\boldsymbol{\phi} = \{\mathbf{W}_{i-1}, \boldsymbol{\theta}_i\}_{i=1}^L$ as feature finding weights in the ResNEst. Obviously, the set of all the weights in the ResNEst is composed of the feature finding weights and prediction weights.

Because $\mathbf{G}_i$ is quite general in the ResNEst, any direct characterization on the landscape of ERM problem seems intractable. Thus, we propose to utilize the basis function modeling point of view in the ResNEst and analyze the following ERM problem:

$$(\mathbf{P}_{\boldsymbol{\phi}}) \min_{\mathbf{W}_L, \mathbf{W}_{L+1}} \mathcal{R}(\mathbf{W}_L, \mathbf{W}_{L+1}; \boldsymbol{\phi}) \tag{3}$$

where $\mathcal{R}(\mathbf{W}_L, \mathbf{W}_{L+1}; \boldsymbol{\phi}) = \frac{1}{N} \sum_{n=1}^N \ell\left(\hat{\mathbf{y}}_{L\text{-ResNEst}}^{\boldsymbol{\phi}}(\mathbf{x}^n), \mathbf{y}^n\right)$ for any fixed feature finding weights $\boldsymbol{\phi}$. We have used $\ell$ and $\{(\mathbf{x}^n, \mathbf{y}^n)\}_{n=1}^N$ to denote the loss function and training data, respectively. $\hat{\mathbf{y}}_{L\text{-ResNEst}}^{\boldsymbol{\phi}}$ denotes a ResNEst using a fixed feature finding weights $\boldsymbol{\phi}$. Although $(\mathbf{P}_{\boldsymbol{\phi}})$ has less optimization variables and looks easier than (P), Proposition 1 shows that it is a non-convex problem. Remark 1 explains why understanding $(\mathbf{P}_{\boldsymbol{\phi}})$ is valuable.

**Remark 1.** *Let the set of all local minimizers of $(\mathbf{P}_{\boldsymbol{\phi}})$ using any possible features equip with the corresponding $\boldsymbol{\phi}$. Then, this set is a superset of the set of all local minimizers of the original ERM problem (P). Any characterization of $(\mathbf{P}_{\boldsymbol{\phi}})$ can then be translated to (P) (see Corollary 2 for example).*

**Assumption 1.** $\sum_{n=1}^N \mathbf{v}_L(\mathbf{x}^n) \mathbf{y}^{nT} \neq \mathbf{0}$ and $\sum_{n=1}^N \mathbf{v}_L(\mathbf{x}^n) \mathbf{v}_L(\mathbf{x}^n)^T$ *is full rank.*

**Proposition 1.** *If $\ell$ is the squared loss and Assumption 1 is satisfied, then (a) the objective function of $(\mathbf{P}_{\boldsymbol{\phi}})$ is non-convex and non-concave; (b) every critical point that is not a local minimizer is a saddle point in $(\mathbf{P}_{\boldsymbol{\phi}})$.*

The proof of Proposition 1 is deferred to Appendix A.1 in the supplementary material. Due to the product $\mathbf{W}_{L+1} \mathbf{W}_L$ in $\mathcal{R}(\mathbf{W}_L, \mathbf{W}_{L+1}; \boldsymbol{\phi})$, our Assumption 1 is similar to one of the important data assumptions used in deep linear networks [Baldi and Hornik, 1989, Kawaguchi, 2016]. Assumption 1 is easy to be satisfied as we can always perturb $\boldsymbol{\phi}$ if the last nonlinear feature and dataset do not fit the assumption. Although Proposition 1 (a) examines the non-convexity for a fixed $\boldsymbol{\phi}$, the result can be extended to the original ERM problem (P) for the ResNEst. That is, if there exists at least one $\boldsymbol{\phi}$ such that Assumption 1 is satisfied, then the objective function for the optimization problem (P) is also non-convex and non-concave because there exists at least one point in the domain at which the Hessian is indefinite. As a result, this non-convex loss landscape in (P) immediately raises issues about suboptimal local minima in the loss landscape. This leads to an important question: Can we guarantee the quality of local minima with respect to some reference models that are known to be good enough?

## 2.3 Finding reference models: bounding empirical risks via augmentation

To avoid the coupling problem in ResNEsts, we propose a new architecture in Figure 1 called augmented ResNEst or A-ResNEst. An $L$-block A-ResNEst introduces another set of parameters $\{\mathbf{H}_i\}_{i=0}^{L}$ to replace every bilinear map on each feature in (2) with a linear map:

$$\hat{\mathbf{y}}_{L\text{-A-ResNEst}}\left(\mathbf{x}\right) = \sum_{i=0}^{L} \mathbf{H}_i \mathbf{v}_i\left(\mathbf{x}\right). \tag{4}$$

Now, the function $\hat{\mathbf{y}}_{L\text{-A-ResNEst}}$ is linear with respect to all the prediction weights $\{\mathbf{H}_i\}_{i=0}^{L}$. Note that the parameters $\{\mathbf{W}_i\}_{i=0}^{L-1}$ still exist and are now dedicated to feature finding. On the other hand, $\mathbf{W}_L$ and $\mathbf{W}_{L+1}$ are deleted since they are not used in the A-ResNEst. As a result, the corresponding ERM problem (PA) is defined on $(\mathbf{H}_0, \cdots, \mathbf{H}_L, \phi)$. We denote $\mathcal{A}$ as the empirical risk in A-ResNEsts. The prediction weights are now different from the ResNEst as the A-ResNEst uses $\{\mathbf{H}_i\}_{i=0}^{L}$. Because any A-ResNEst prevents the coupling problem, it exhibits a nice property shown below.

**Assumption 2.** *The loss function $\ell(\hat{\mathbf{y}}, \mathbf{y})$ is differentiable and convex in $\hat{\mathbf{y}}$ for any $\mathbf{y}$.*

**Proposition 2.** *Let $\left(\mathbf{H}_0^*, \cdots, \mathbf{H}_L^*\right)$ be any local minimizer of the following optimization problem:*

$$(PA_\phi) \quad \min_{\mathbf{H}_0, \cdots, \mathbf{H}_L} \mathcal{A}\left(\mathbf{H}_0, \cdots, \mathbf{H}_L; \phi\right) \tag{5}$$

*where $\mathcal{A}\left(\mathbf{H}_0, \cdots, \mathbf{H}_L; \phi\right) = \frac{1}{N} \sum_{n=1}^{N} \ell\left(\hat{\mathbf{y}}_{L\text{-A-ResNEst}}^{\phi}\left(\mathbf{x}^n\right), \mathbf{y}^n\right)$. If Assumption 2 is satisfied, then the optimization problem in (5) is convex and*

$$\epsilon\left(\mathbf{W}_L^*, \mathbf{W}_{L+1}^*; \phi\right) = \mathcal{R}\left(\mathbf{W}_L^*, \mathbf{W}_{L+1}^*; \phi\right) - \mathcal{A}\left(\mathbf{H}_0^*, \cdots, \mathbf{H}_L^*; \phi\right) \geq 0 \tag{6}$$

*for any local minimizer $\left(\mathbf{W}_L^*, \mathbf{W}_{L+1}^*\right)$ of $(P_\phi)$ using arbitrary feature finding parameters $\phi$.*

The proof of Proposition 2 is deferred to Appendix A.2 in the supplementary material. According to Proposition 2, A-ResNEst establishes empirical risk lower bounds (ERLBs) for a ResNEst. Hence, for the same $\phi$ picked arbitrarily, an A-ResNEst is better than a ResNEst in terms of any pair of two local minima in their loss landscapes. Assumption 2 is practical because it is satisfied for two commonly used loss functions in regression and classification, i.e., the squared loss and cross-entropy loss. Other losses such as the logistic loss and smoothed hinge loss also satisfy this assumption.

## 2.4 Necessary condition for strictly improved residual representations

What properties are fundamentally required for features to be good, i.e., able to strictly improve the residual representation over blocks? With A-ResNEsts, we are able to straightforwardly answer this question. A fundamental answer is they need to be at least *linearly unpredictable*. Note that $\mathbf{v}_i$ must be linearly unpredictable by $\mathbf{v}_0, \cdots, \mathbf{v}_{i-1}$ if

$$\mathcal{A}\left(\mathbf{H}_0^*, \mathbf{H}_1^*, \cdots, \mathbf{H}_{i-1}^*, \mathbf{0}, \cdots, \mathbf{0}, \phi^*\right) > \mathcal{A}\left(\mathbf{H}_0^*, \mathbf{H}_1^*, \cdots, \mathbf{H}_i^*, \mathbf{0}, \cdots, \mathbf{0}, \phi^*\right) \tag{7}$$

for any local minimum $\left(\mathbf{H}_0^*, \cdots, \mathbf{H}_L^*, \phi^*\right)$ in (PA). In other words, the residual representation $\mathbf{x}_i$ is not strictly improved from the previous representation $\mathbf{x}_{i-1}$ if the feature $\mathbf{v}_i$ is linearly predictable by the previous features. Fortunately, the linearly unpredictability of $\mathbf{v}_i$ is usually satisfied when $\mathbf{G}_i$ is nonlinear; and the set of features can be viewed as a basis function. This viewpoint also suggests avenues for improving feature construction through imposition of various constraints. By Proposition 2, the relation in (7) always holds with equality, i.e., the residual representation $\mathbf{x}_i$ is guaranteed to be always no worse than the previous one $\mathbf{x}_{i-1}$ at any local minimizer obtained from an A-ResNEst.

## 3 Wide ResNEsts with bottleneck residual blocks always attain ERLBs

**Assumption 3.** $M \geq N_o$.

**Assumption 4.** *The linear inverse problem $\mathbf{x}_{L-1} = \sum_{i=0}^{L-1} \mathbf{W}_i \mathbf{v}_i$ has a unique solution.*

**Theorem 1.** *If Assumption 2 and 3 are satisfied, then the following two properties are true in $(P_\phi)$ under any $\phi$ such that Assumption 4 holds: (a) every critical point with full rank $\mathbf{W}_{L+1}$ is a global minimizer; (b) $\epsilon = 0$ for every local minimizer.*

The proof of Theorem 1 is deferred to Appendix A.3 in the supplementary material. Theorem 1 (a) provides a sufficient condition for a critical point to be a global minimum of ($P_\phi$). Theorem 1 (b) gives an affirmative answer for every local minimum in ($P_\phi$) to attain the ERLB. To be more specific, any pair of obtained local minima from the ResNEst and the A-ResNEst using the same arbitrary $\phi$ are equally good. In addition, the implication of Theorem 1 (b) is that every local minimum of ($P_\phi$) is also a global minimum despite its non-convex landscape (Proposition 1), which suggests there exists no suboptimal local minimum for the optimization problem ($P_\phi$). One can also establish the same results for local minimizers of (P) under the same set of assumptions by replacing "($P_\phi$) under any $\phi$" with just "(P)" in Theorem 1. Such a modification may gain more clarity, but is more restricted than the original statement due to Remark 1. Note that Theorem 1 is not limited to fixing any weights during training; and it applies to both normal training (train all the weights in a network as a whole) and blockwise or layerwise training procedures.

## 3.1 Improved representation guarantees

By Remark 1 and Theorem 1 (b), we can then establish the following representational guarantee.

**Remark 2.** *Let Assumption 2 and 3 be true. Any local minimizer of (P) such that Assumption 4 is satisfied guarantees (a) monotonically improved (no worse) residual representations over blocks; (b) every residual representation is better than the input representation in the linear prediction sense.*

Although there may exist suboptimal local minima in the optimization problem (P), Remark 2 suggests that such minima still improve residual representations over blocks under practical conditions. Mathematically, Remark 2 (a) and Remark 2 (b) are described by Corollary 1 and the general version of Corollary 2, respectively. Corollary 1 compares the minimum empirical risk obtained at any two representations among $\mathbf{x}_1$ to $\mathbf{x}_L$ for any given network satisfying the assumptions; and Corollary 2 extends this comparison to the input representation.

**Corollary 1.** *Let Assumption 2 and 3 be true. Any local minimum of ($P_\alpha$) is smaller than or equal to any local minimum of ($P_\beta$) under Assumption 4 for any $\alpha = \{\mathbf{W}_{i-1}, \boldsymbol{\theta}_i\}_{i=1}^{L_\alpha}$ and $\beta = \{\mathbf{W}_{i-1}, \boldsymbol{\theta}_i\}_{i=1}^{L_\beta}$ where $L_\alpha$ and $L_\beta$ are positive integers such that $L_\alpha > L_\beta$.*

The proof of Corollary 1 is deferred to Appendix A.4 in the supplementary material. Because Corollary 1 holds true for any properly given weights, one can apply Corollary 1 to proper local minimizers of (P). Corollary 2 ensures that ResNEsts are guaranteed to be no worse than the best linear predictor under practical assumptions. This property is useful because linear estimators are widely used in signal processing applications and they can now be confidently replaced with ResNEsts.

**Corollary 2.** *Let $\left(\mathbf{W}_0^*, \cdots, \mathbf{W}_{L+1}^*, \boldsymbol{\theta}_1^*, \cdots, \boldsymbol{\theta}_L^*\right)$ be any local minimizer of (P) and $\phi^* = \{\mathbf{W}_{i-1}^*, \boldsymbol{\theta}_i^*\}_{i=1}^L$. If Assumption 2, 3 and 4 are satisfied, then (a) $\mathcal{R}\left(\mathbf{W}_0^*, \cdots, \mathbf{W}_{L+1}^*, \boldsymbol{\theta}_1^*, \cdots, \boldsymbol{\theta}_L^*\right) \leq \min_{\mathbf{A} \in \mathbb{R}^{N_o \times N_{in}}} \frac{1}{N} \sum_{n=1}^N \ell\left(\mathbf{A}\mathbf{x}^n, \mathbf{y}^n\right)$; (b) the above inequality is strict if $\mathcal{A}\left(\mathbf{H}_0^*, \mathbf{0}, \cdots, \mathbf{0}, \phi^*\right) > \mathcal{A}\left(\mathbf{H}_0^*, \cdots, \mathbf{H}_L^*, \phi^*\right)$.*

The proof of Corollary 2 is deferred to Appendix A.5 in the supplementary material. To the best of our knowledge, Corollary 2 is the first theoretical guarantee for vector-valued ResNet-like models that have arbitrary residual blocks to outperform any linear predictors. Corollary 2 is more general than the results in [Shamir, 2018, Kawaguchi and Bengio, 2019, Yun et al., 2019] because it is not limited to assumptions like scalar-valued output or single residual block. In fact, we can have a even more general statement because any local minimum obtained from ($P_\phi$) with random or any $\phi$ is better than the minimum empirical risk provided by the best linear predictor, under the same assumptions used in Corollary 2. This general version fully describes Remark 2 (b).

Theorem 1, Corollary 1 and Corollary 2 are quite general because they are not limited to specific loss functions, residual functions, or datasets. Note that we do not impose any assumptions such as differentiability or convexity on the neural network $\mathbf{G}_i$ for $i = 1, 2, \cdots, L$ in residual functions. Assumption 3 is practical because the expansion factor $M$ is usually larger than the input dimension $N_{in}$; and the output dimension $N_o$ is usually not larger than the input dimension for most supervised learning tasks using sensory input. Assumption 4 states that the features need to be uniquely invertible from the residual representation. Although such an assumption requires a special architectural design, we find that it is always satisfied empirically after random initialization or training when the "bottleneck condition" is satisfied.

## 3.2 How to design architectures with representational guarantees?

Notice that one must be careful with the ResNEst architectural design so as to enjoy Theorem 1, Corollary 1 and Corollary 2. A ResNEst needs to be wide enough such that $M \geq \sum_{i=0}^{L-1} K_i$ to necessarily satisfy Assumption 4. We call such a sufficient condition on the width and feature dimensionalities as a *bottleneck condition*. Because each nonlinear feature size $K_i$ for $i < L$ (say $L > 1$) must be smaller than the dimensionality of the residual representation $M$, each of these residual functions is a *bottleneck design* [He et al., 2016a,b, Zagoruyko and Komodakis, 2016] forming a bottleneck residual block. We now explicitly see the importance of the expansion layer. Without the expansion, the dimenionality of the residual representation is limited to the input dimension. As a result, Assumption 4 cannot be satisfied for $L > 1$; and the analysis for the ResNEst with multiple residual blocks remains intractable or requires additional assumptions on residual functions.

Loosely speaking, a sufficiently wide expansion or satisfaction of the bottleneck condition implies Assumption 4. If the bottleneck condition is satisfied, then ResNEsts are equivalent to A-ResNEsts for a given $\phi$, i.e., $\epsilon = 0$. If not (e.g., basic blocks are used in a ResNEst), then a ResNEst can have a problem of diminishing feature reuse or end up with poor performance even though it has excellent features that can be fully exploited by an A-ResNEst to yield better performance, i.e., $\epsilon > 0$. From such a viewpoint, Theorem 1 supports the empirical findings in [He et al., 2016a] that bottleneck blocks are more economical than basic blocks. Our results thus recommend A-ResNEsts over ResNEsts if the bottleneck condition cannot be satisfied.

### 3.3 Guarantees on saddle points

In addition to guarantees for the quality of local minima, we find that ResNEsts can easily escape from saddle points due to the nice property shown below.

**Theorem 2.** *If $\ell$ is the squared loss, and Assumption 1 and 3 are satisfied, then the following two properties are true at every saddle point of ($P_\phi$) under any $\phi$ such that Assumption 4 holds: (a) $\mathbf{W}_{L+1}$ is rank-deficient; (b) there exists at least one direction with strictly negative curvature.*

The proof of Theorem 2 is deferred to Appendix A.6 in the supplementary material. In contrast to Theorem 1 (a), Theorem 2 (a) provides a necessary condition for a saddle point. Although ($P_\phi$) is a non-convex optimization problem according to Proposition 1 (a), Theorem 2 (b) suggests a desirable property for saddle points in the loss landscape. Because there exists at least one direction with strictly negative curvature at every saddle point that satisfies the bottleneck condition, the second-order optimization methods can rapidly escape from saddle points [Dauphin et al., 2014]. If the first-order methods are used, the randomness in stochastic gradient helps the first-order methods to escape from the saddle points [Ge et al., 2015]. Again, we require the bottleneck condition to be satisfied in order to guarantee such a nice property about saddle points. Note that Theorem 2 is not limited to fixing any weights during training; and it applies to both normal training and blockwise training procedures due to Remark 1.

## 4 DenseNEsts are wide ResNEsts with bottleneck residual blocks equipped with orthogonalities

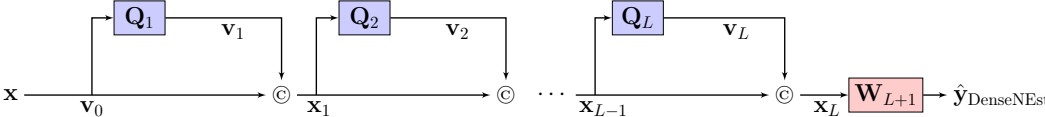

Figure 3: A generic vector-valued DenseNEst that has a chain of $L$ dense blocks (or units). The symbol "©" represents the concatenation operation. We intentionally draw a DenseNEst in such a form to emphasize its relationship to a ResNEst (see Proposition 4).

Instead of adding one nonlinear feature in each block and remaining in same space $\mathbb{R}^M$, the DenseNEst model shown in Figure 3 preserves each of features in their own subspaces by a sequential concatenation at each block. For an $L$-block DenseNEst, we define the $i$-th dense block as a function $\mathbb{R}^{M_{i-1}} \mapsto \mathbb{R}^{M_i}$ of the form

$$\mathbf{x}_i = \mathbf{x}_{i-1} © \mathbf{Q}_i \left( \mathbf{x}_{i-1}; \boldsymbol{\theta}_i \right) \tag{8}$$

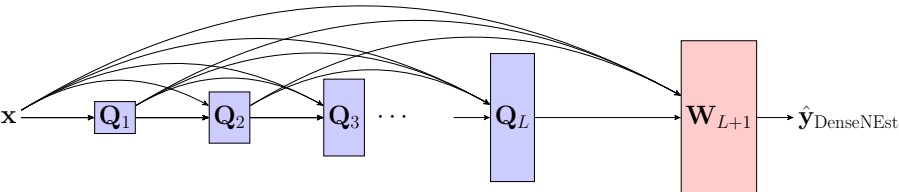

Figure 4: An equivalence to Figure 3 emphasizing the growth of the input dimension at each block.

for $i = 1, 2, \cdots, L$ where the dense function $\mathbf{Q}_i$ is a general nonlinear function; and $\mathbf{x}_i$ is the output of the $i$-th dense block. The symbol © concatenates vector $\mathbf{x}_{i-1}$ and vector $\mathbf{Q}_i(\mathbf{x}_{i-1}; \boldsymbol{\theta}_i)$ and produces a higher-dimensional vector $\begin{bmatrix} \mathbf{x}_{i-1}^T & \mathbf{Q}_i(\mathbf{x}_{i-1}; \boldsymbol{\theta}_i)^T \end{bmatrix}^T$. We define $\mathbf{x}_0 = \mathbf{x}$ where $\mathbf{x} \in \mathbb{R}^{N_{in}}$ is the input to the DenseNEst. For all $i \in \{1, 2, \cdots, L\}$, $\mathbf{Q}_i(\mathbf{x}_{i-1}; \boldsymbol{\theta}_i) : \mathbb{R}^{M_{i-1}} \mapsto \mathbb{R}^{D_i}$ is a function implemented by a neural network with parameters $\boldsymbol{\theta}_i$ where $D_i = M_i - M_{i-1} \geq 1$ with $M_0 = N_{in} = D_0$. The output of a DenseNEst is defined as $\hat{\mathbf{y}}_{\text{DenseNEst}} = \mathbf{W}_{L+1}\mathbf{x}_L$ for $\mathbf{W}_{L+1} \in \mathbb{R}^{N_o \times M_L}$, which can be written as

$$\mathbf{W}_{L+1}\left(\mathbf{x}_0 © \mathbf{Q}_1(\mathbf{x}_0; \boldsymbol{\theta}_1) © \cdots © \mathbf{Q}_L(\mathbf{x}_{L-1}; \boldsymbol{\theta}_L)\right) = \sum_{i=0}^{L} \mathbf{W}_{L+1,i}\mathbf{v}_i(\mathbf{x}) \tag{9}$$

where $\mathbf{v}_i(\mathbf{x}) = \mathbf{Q}_i(\mathbf{x}_{i-1}; \boldsymbol{\theta}_i) = \mathbf{Q}_i(\mathbf{x}_0 © \mathbf{v}_1 © \mathbf{v}_2 © \cdots © \mathbf{v}_{i-1}; \boldsymbol{\theta}_i)$ for $i = 1, 2, \cdots, L$ are regarded as nonlinear features of the input $\mathbf{x}$. We define $\mathbf{v}_0 = \mathbf{x}$ as the linear feature. $\mathbf{W}_{L+1} = \begin{bmatrix} \mathbf{W}_{L+1,0} & \mathbf{W}_{L+1,1} & \cdots & \mathbf{W}_{L+1,L} \end{bmatrix}$ is the prediction weight matrix in the DenseNEst as all the weights which are responsible for the prediction is in this single matrix from the viewpoint of basis function modeling. The ERM problem (PD) for the DenseNEst is defined on $(\mathbf{W}_{L+1}, \boldsymbol{\theta}_1, \cdots, \boldsymbol{\theta}_L)$. To fix the features, the set of parameters $\phi = \{\boldsymbol{\theta}_i\}_{i=1}^{L}$ needs to be fixed. Therefore, the DenseNEst ERM problem for any fixed features, denoted as (PD$_\phi$), is fairly straightforward as it only requires to optimize over a single weight matrix, i.e.,

$$(\text{PD}_\phi) \min_{\mathbf{W}_{L+1}} \mathcal{D}(\mathbf{W}_{L+1}; \phi) \tag{10}$$

where $\mathcal{D}(\mathbf{W}_{L+1}; \phi) = \frac{1}{N}\sum_{n=1}^{N} \ell\left(\hat{\mathbf{y}}_{L\text{-DenseNEst}}^{\phi}(\mathbf{x}^n), \mathbf{y}^n\right)$. Unlike ResNEsts, there is no such coupling between the feature finding and linear prediction in DenseNEsts. Compared to ResNEsts or A-ResNEsts, the way the features are generated in DenseNEsts generally makes the linear predictability even more unlikely. To see that, note that the $\mathbf{Q}_i$ directly applies on the concatenation of all previous features; however, the $\mathbf{G}_i$ applies on the sum of all previous features.

Different from a ResNEst which requires Assumption 2, 3 and 4 to guarantee its superiority with respect to the best linear predictor (Corollary 2), the corresponding guarantee in a DenseNEst shown in Proposition 3 requires weaker assumptions.

**Proposition 3.** *If Assumption 2 is satisfied, then any local minimum of (PD) is smaller than or equal to the minimum empirical risk given by any linear predictor of the input.*

The proof of Proposition 3 is deferred to Appendix A.7 in the supplementary material. Notice that no special architectural design in a DenseNEst is required to make sure it always outperforms the best linear predictor. Any DenseNEst is always better than any linear predictor when the loss function is differentiable and convex (Assumption 2). Such an advantage can be explained by the $\mathbf{W}_{L+1}$ in the DenseNEst. Because $\mathbf{W}_{L+1}$ is the only prediction weight matrix which is directly applied onto the concatenation of all the features, (PD$_\phi$) is a convex optimization problem. We point out the difference of $\mathbf{W}_{L+1}$ between the ResNEst and DenseNEst. In the ResNEst, $\mathbf{W}_{L+1}$ needs to interpret the features from the residual representation; while the $\mathbf{W}_{L+1}$ in the DenseNEst directly accesses the features. That is why we require Assumption 4 in the ResNEst to eliminate any ambiguity on the feature interpretation.

Can a ResNEst and a DenseNEst be equivalent? Yes, Proposition 4 establishes a link between them.

**Proposition 4.** *Given any DenseNEst $\hat{\mathbf{y}}_{L\text{-DenseNEst}}$, there exists a wide ResNEst with bottleneck residual blocks $\hat{\mathbf{y}}_{L\text{-ResNEst}}^{\phi}$ such that $\hat{\mathbf{y}}_{L\text{-ResNEst}}^{\phi}(\mathbf{x}) = \hat{\mathbf{y}}_{L\text{-DenseNEst}}(\mathbf{x})$ for all $\mathbf{x} \in \mathbb{R}^{N_{in}}$. If, in addition, Assumption 2 and 3 are satisfied, then $\epsilon = 0$ for every local minimizer of (P$_\phi$).*

The proof of Proposition 4 is deferred to Appendix A.8 in the supplementary material. Because the concatenation of two given vectors can be represented by an addition over two vectors projected onto a higher dimensional space with disjoint supports, one straightforward construction for an equivalent ResNEst is to sufficiently expand the input space and enforce the orthogonality of all the column vectors in $\mathbf{W}_0, \mathbf{W}_1, \cdots, \mathbf{W}_L$. As a result, any DenseNEst can be viewed as a ResNEst that always satisfies Assumption 4 and of course the bottleneck condition no matter how we train the DenseNEst or select its hyperparameters, leading to the desirable guarantee, i.e., any local minimum obtained in optimizing the prediction weights of the resulting ResNEst from any DenseNEst always attains the lower bound. Thus, DenseNEsts are certified as being advantageous over ResNEsts by Proposition 4. For example, a small $M$ may be chosen and then the guarantee in Theorem 1 can no longer exist, i.e., $\epsilon > 0$. However, the corresponding ResNEst induced by a DenseNEst always achieves $\epsilon = 0$. Hence, Proposition 4 can be regarded as a theoretical support for why standard DenseNets [Huang et al., 2017] are in general better than standard ResNets [He et al., 2016b].

## 5 Related work

In this section, we discuss ResNet works that investigate on properties of local minima and give more details for our important references that appear in the introduction. We focus on highlighting their results and assumptions used so as to compare to our theoretical results derived from practical assumptions. The earliest theoretical work for ResNets can be dated back to [Hardt and Ma, 2017] which proved a vector-valued ResNet-like model using a linear residual function in each residual block has no spurious local minima (local minima that give larger objective values than the global minima) under squared loss and near-identity region assumptions. There are results [Li and Yuan, 2017, Liu et al., 2019] proved that stochastic gradient descent can converge to the global minimum in scalar-valued two-layer ResNet-like models; however, such a desirable property relies on strong assumptions including single residual block and Gaussian input distribution. Li et al. [2018] visualized the loss landscapes of a ResNet and its plain counterpart (without skip connections); and they showed that the skip connections promote flat minimizers and prevent the transition to chaotic behavior. Liang et al. [2018] showed that scalar-valued and single residual block ResNet-like models can have zero training error at all local minima by making strong assumptions in the data distribution and loss function for a binary classification problem. In stead of pursuing local minima are global in the empirical risk landscape using strong assumptions, Shamir [2018] first took a different route and proved that a scalar-valued ResNet-like model with a direct skip connection from input to output layer (single residual block) is better than any linear predictor under mild assumptions. To be more specific, he showed that every local minimum obtained in his model is no worse than the global minimum in any linear predictor under more generalized residual functions and no assumptions on the data distribution. He also pointed out that the analysis for the vector-valued case is nontrivial. Kawaguchi and Bengio [2019] overcame such a difficulty and proved that vector-valued models with single residual block is better than any linear predictor under weaker assumptions. Yun et al. [2019] extended the prior work by Shamir [2018] to multiple residual blocks. Although the model considered is closer to a standard ResNet compared to previous works, the model output is assumed to be scalar-valued. All above-mentioned works do not take the first layer that appears before the first residual block in standard ResNets into account. As a result, the dimensionality of the residual representation in their simplified ResNet models is constrained to be the same size as the input.

## Broader impact

One of the mysteries in ResNets and DenseNets is that learning better DNN models seems to be as easy as stacking more blocks. In this paper, we define three generalized and analyzable DNN architectures, i.e., ResNEsts, A-ResNEsts, and DenseNEsts, to answer this question. Our results not only establish guarantees for monotonically improved representations over blocks, but also assure that all linear (affine) estimators can be replaced by our architectures without harming performance. We anticipate these models can be friendly options for researchers or engineers who value or mostly rely on linear estimators or performance guarantees in their problems. In fact, these models should yield much better performance as they can be viewed as basis function models with data-driven bases that guarantee to be always better than the best linear estimator. Our contributions advance the fundamental understanding of ResNets and DenseNets, and promote their use cases through a certificate of attractive guarantees.

## Acknowledgments and disclosure of funding

We would like to thank the anonymous reviewers for their constructive comments. This work was supported in part by NSF under Grant CCF-2124929 and Grant IIS-1838830, in part by NIH/NIDCD under Grant R01DC015436, Grant R21DC015046, and Grant R33DC015046, in part by Halıcıoğlu Data Science Institute, and in part by Wrethinking, the Foundation.

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
