# A  Proofs

## A.1  Proof of Proposition 1

*Proof.* Let

$$\mathbf{V}_i = \begin{bmatrix} \mathbf{v}_i(\mathbf{x}^1) & \mathbf{v}_i(\mathbf{x}^2) & \cdots & \mathbf{v}_i(\mathbf{x}^N) \end{bmatrix} \tag{11}$$

for $i = 0, 1, \cdots, L$ and

$$\mathbf{\Delta} = \left( \mathbf{W}_{L+1} \sum_{i=0}^{L} \mathbf{W}_i \mathbf{V}_i - \mathbf{Y} \right)^T = \left( \hat{\mathbf{Y}} - \mathbf{Y} \right)^T = \begin{bmatrix} \boldsymbol{\delta}_1 & \boldsymbol{\delta}_2 & \cdots & \boldsymbol{\delta}_{N_o} \end{bmatrix} \tag{12}$$

where $\mathbf{Y} = \begin{bmatrix} \mathbf{y}^1 & \mathbf{y}^2 & \cdots & \mathbf{y}^N \end{bmatrix}$. The Hessian of $\mathcal{R}\left(\mathbf{W}_L, \mathbf{W}_{L+1}; \boldsymbol{\phi}\right)$ in (P$_\phi$) is given by

$$
\begin{aligned}
\nabla^2 \mathcal{R} &= \begin{bmatrix} \frac{\partial^2 \mathcal{R}}{\partial \mathrm{vec}\left(\mathbf{W}_L^T\right)^2} & \frac{\partial^2 \mathcal{R}}{\partial \mathrm{vec}\left(\mathbf{W}_{L+1}^T\right) \partial \mathrm{vec}\left(\mathbf{W}_L^T\right)} \\ \frac{\partial^2 \mathcal{R}}{\partial \mathrm{vec}\left(\mathbf{W}_L^T\right) \partial \mathrm{vec}\left(\mathbf{W}_{L+1}^T\right)} & \frac{\partial^2 \mathcal{R}}{\partial \mathrm{vec}\left(\mathbf{W}_{L+1}^T\right)^2} \end{bmatrix} \\
&= \frac{2}{N} \begin{bmatrix} \mathbf{W}_{L+1}^T \mathbf{W}_{L+1} \otimes \mathbf{V}_L \mathbf{V}_L^T & \mathbf{W}_{L+1}^T \otimes \mathbf{V}_L \sum_{i=0}^{L} \mathbf{V}_i^T \mathbf{W}_i^T + \mathbf{E} \\ \mathbf{W}_{L+1} \otimes \sum_{i=0}^{L} \mathbf{W}_i \mathbf{V}_i \mathbf{V}_L^T + \mathbf{E}^T & \mathbf{I}_{N_o} \otimes \sum_{i=0}^{L} \mathbf{W}_i \mathbf{V}_i \left( \sum_{i=0}^{L} \mathbf{W}_i \mathbf{V}_i \right)^T \end{bmatrix}
\end{aligned} \tag{13}
$$

where

$$\mathbf{E} = \begin{bmatrix} \mathbf{I}_M \otimes \mathbf{V}_L \boldsymbol{\delta}_1 & \cdots & \mathbf{I}_M \otimes \mathbf{V}_L \boldsymbol{\delta}_{N_o} \end{bmatrix}. \tag{14}$$

We have used $\otimes$ to denote the Kronecker product. See Appendix A.9 for the derivation of the Hessian. By the generalized Schur complement,

$$\nabla^2 \mathcal{R} \succeq \mathbf{0} \implies \mathrm{range}\left( \frac{\partial^2 \mathcal{R}}{\partial \mathrm{vec}\left(\mathbf{W}_{L+1}^T\right) \partial \mathrm{vec}\left(\mathbf{W}_L^T\right)} \right) \subseteq \mathrm{range}\left( \frac{\partial^2 \mathcal{R}}{\partial \mathrm{vec}\left(\mathbf{W}_L^T\right)^2} \right) \tag{15}$$

which implies the projection of $\frac{\partial^2 \mathcal{R}}{\partial \mathrm{vec}\left(\mathbf{W}_{L+1}^T\right) \partial \mathrm{vec}\left(\mathbf{W}_L^T\right)}$ onto the range of $\frac{\partial^2 \mathcal{R}}{\partial \mathrm{vec}\left(\mathbf{W}_L^T\right)^2}$ is itself. As a result,

$$\left( \mathbf{I}_{MK_L} - \frac{\partial^2 \mathcal{R}}{\partial^2 \mathrm{vec}\left(\mathbf{W}_L^T\right)} \left( \frac{\partial^2 \mathcal{R}}{\partial^2 \mathrm{vec}\left(\mathbf{W}_L^T\right)} \right)^{\dagger} \right) \frac{\partial^2 \mathcal{R}}{\partial \mathrm{vec}\left(\mathbf{W}_{L+1}^T\right) \partial \mathrm{vec}\left(\mathbf{W}_L^T\right)} = \mathbf{0} \tag{16}$$

where $\dagger$ denotes the Moore-Penrose pseudoinverse. Substituting the submatrices in (13) to the above equation, we obtain

$$\frac{2}{N} \begin{bmatrix} \left( \left( \mathbf{I}_M - \mathbf{W}_{L+1}^T \mathbf{W}_{L+1} \left( \mathbf{W}_{L+1}^T \mathbf{W}_{L+1} \right)^{\dagger} \right)^T \otimes \boldsymbol{\delta}_1^T \mathbf{V}_L^T \right)^T \\ \vdots \\ \left( \left( \mathbf{I}_M - \mathbf{W}_{L+1}^T \mathbf{W}_{L+1} \left( \mathbf{W}_{L+1}^T \mathbf{W}_{L+1} \right)^{\dagger} \right)^T \otimes \boldsymbol{\delta}_{N_o}^T \mathbf{V}_L^T \right) \end{bmatrix} = \mathbf{0} \tag{17}$$

which implies

$$\mathbf{W}_{L+1}^T \mathbf{W}_{L+1} \left( \mathbf{W}_{L+1}^T \mathbf{W}_{L+1} \right)^{\dagger} = \mathbf{I}_M \quad \text{or} \quad \mathbf{V}_L \mathbf{\Delta} = \mathbf{0}. \tag{18}$$

On the other hand, the above condition is also necessary for the Hessian to be negative semidefinite because $\nabla^2 \mathcal{R} \preceq \mathbf{0} \implies -\nabla^2 \mathcal{R} \succeq \mathbf{0}$ which implies (16).

Now, using the assumption $\sum_{n=1}^{N} \mathbf{v}_L\left(\mathbf{x}^n\right) \mathbf{y}^{nT} \neq \mathbf{0}$, notice that the condition in (18) is not satisfied for any point in the set

$$\mathcal{S} = \left\{ \left(\mathbf{W}_L, \mathbf{W}_{L+1}\right) \middle| \mathbf{W}_L \in \mathbb{R}^{M \times K_L}, \mathbf{W}_{L+1} = \mathbf{0} \right\}. \tag{19}$$

Hence, there exist some points in the domain at which the Hessian is indefinite. The objective function $\mathcal{R}\left(\mathbf{W}_L, \mathbf{W}_{L+1}; \phi\right)$ in $(\mathrm{P}_\phi)$ is non-convex and non-concave. We have proved the statement (a).

By the generalized Schur complement and the assumption that $\sum_{n=1}^N \mathbf{v}_L\left(\mathbf{x}^n\right) \mathbf{v}_L\left(\mathbf{x}^n\right)^T$ is full rank, we have

$$\nabla^2 \mathcal{R} \preceq \mathbf{0} \implies \frac{\partial^2 \mathcal{R}}{\partial \mathrm{vec}\left(\mathbf{W}_L^T\right)^2} \preceq \mathbf{0} \implies \mathbf{W}_{L+1} = \mathbf{0} \tag{20}$$

where we have used the spectrum property of the Kronecker product and the positive definiteness of $\mathbf{V}_L \mathbf{V}_L^T$. Notice that this is a contradiction because any point with $\mathbf{W}_{L+1} = \mathbf{0}$ is in the set $\mathcal{S}$. Hence, there exists no point at which the Hessian is negative semidefinite. Because the negative semidefiniteness is a necessary condition for a local maximum, every critical point is then either a local minimum or a saddle point. We have proved the statement (b). $\qquad\square$

## A.2 Proof of Proposition 2

*Proof.* $\mathcal{A}\left(\mathbf{H}_0, \cdots, \mathbf{H}_L; \phi\right)$ is convex in $\begin{bmatrix} \mathbf{H}_0 & \mathbf{H}_1 & \cdots & \mathbf{H}_L \end{bmatrix}$ because it is a nonnegative weighted sum of convex functions composited with affine mappings. Thus, $(\mathrm{PA}_\phi)$ is a convex optimization problem and $\left(\mathbf{H}_0^*, \cdots, \mathbf{H}_L^*\right)$ is the best linear fit using $\phi$. That is, for any local minimizer $\left(\mathbf{H}_0^*, \cdots, \mathbf{H}_L^*\right)$, it is always true that

$$\frac{1}{N} \sum_{n=1}^N \ell\left(\sum_{i=0}^L \mathbf{H}_i^* \mathbf{v}_i(\mathbf{x}^n), \mathbf{y}^n\right) \leq \frac{1}{N} \sum_{n=1}^N \ell\left(\sum_{i=0}^L \mathbf{A}_i \mathbf{v}_i(\mathbf{x}^n), \mathbf{y}^n\right) \tag{21}$$

for arbitrary $\mathbf{A}_i \in \mathbb{R}^{N_o \times K_i}, i = 0, 1, \cdots, L$. $\qquad\square$

## A.3 Proof of Theorem 1

*Proof.* By the convexity in Proposition 2, every critical point in $(\mathrm{PA}_\phi)$ is a global minimizer. Since the objective function of $(\mathrm{PA}_\phi)$ is differentiable, the first-order derivative is a zero row vector at any critical point, i.e.,

$$
\begin{aligned}
\frac{\partial \mathcal{A}}{\partial \mathrm{vec}\left(\mathbf{H}_i\right)} &= \frac{1}{N} \sum_{n=1}^N \frac{\partial \ell\left(\hat{\mathbf{y}}, \mathbf{y}^n\right)}{\partial \mathrm{vec}\left(\mathbf{H}_i\right)}\Bigg|_{\hat{\mathbf{y}} = \sum_{i=0}^L \mathbf{H}_i \mathbf{v}_i(\mathbf{x}^n)} \\
&= \frac{1}{N} \sum_{n=1}^N \frac{\partial \ell\left(\hat{\mathbf{y}}, \mathbf{y}^n\right)}{\partial \hat{\mathbf{y}}} \frac{\partial \hat{\mathbf{y}}}{\partial \mathrm{vec}\left(\mathbf{H}_i\right)}\Bigg|_{\hat{\mathbf{y}} = \sum_{i=0}^L \mathbf{H}_i \mathbf{v}_i(\mathbf{x}^n)} \\
&= \frac{1}{N} \sum_{n=1}^N \frac{\partial \ell\left(\hat{\mathbf{y}}, \mathbf{y}^n\right)}{\partial \hat{\mathbf{y}}}\Bigg|_{\hat{\mathbf{y}} = \sum_{i=0}^L \mathbf{H}_i \mathbf{v}_i(\mathbf{x}^n)} \left(\mathbf{v}_i\left(\mathbf{x}^n\right)^T \otimes \mathbf{I}_{N_o}\right) \\
&= \frac{1}{N} \sum_{n=1}^N \left(\left(\mathbf{v}_i\left(\mathbf{x}^n\right) \otimes \mathbf{I}_{N_o}\right) \underbrace{\frac{\partial \ell\left(\hat{\mathbf{y}}, \mathbf{y}^n\right)^T}{\partial \hat{\mathbf{y}}}\Bigg|_{\hat{\mathbf{y}} = \sum_{i=0}^L \mathbf{H}_i \mathbf{v}_i(\mathbf{x}^n)}}_{\mathbf{g}_a(\mathbf{x}^n)}\right)^T \\
&= \frac{1}{N} \sum_{n=1}^N \mathrm{vec}\left(\mathbf{g}_a\left(\mathbf{x}^n\right) \mathbf{v}_i\left(\mathbf{x}^n\right)^T\right)^T \\
&= \mathbf{0}
\end{aligned}
\tag{22}
$$

for $i = 0, 1, \cdots, L$. Again, we have used $\otimes$ to denote the Kronecker product. According to (22), the point $\left(\mathbf{H}_0^*, \cdots, \mathbf{H}_L^*\right)$ is a global minimizer in $(\mathrm{PA}_\phi)$ if and only if the sum of rank one matrices is a zero matrix for $i = 0, 1, \cdots, L$, i.e.,

$$\sum_{n=1}^N \mathbf{v}_i\left(\mathbf{x}^n\right) \mathbf{g}_a\left(\mathbf{x}^n\right)^T = \mathbf{0}, \quad i = 0, 1, \cdots, L. \tag{23}$$

Next, we show that every local minimizer $\left(\mathbf{W}_L^*, \mathbf{W}_{L+1}^*\right)$ of $(\mathrm{P}_\phi)$ establishes a corresponding global minimizer $\left(\mathbf{H}_0^*, \cdots, \mathbf{H}_L^*\right)$ in $(\mathrm{PA}_\phi)$ such that $\mathbf{H}_i^* = \mathbf{W}_{L+1}^* \mathbf{W}_i$ for $i = 0, 1, \cdots, L$.

At any local minimizer of $(\mathrm{P}_\phi)$, the first-order necessary condition with respect to $\mathbf{W}_L$ is given by

$$
\begin{aligned}
\frac{\partial \mathcal{R}}{\partial \operatorname{vec}\left(\mathbf{W}_L\right)} &= \frac{1}{N} \sum_{n=1}^{N} \left.\frac{\partial \ell\left(\hat{\mathbf{y}}, \mathbf{y}^n\right)}{\partial \operatorname{vec}\left(\mathbf{W}_L\right)}\right|_{\hat{\mathbf{y}}=\mathbf{W}_{L+1} \sum_{i=0}^{L} \mathbf{W}_i \mathbf{v}_i\left(\mathbf{x}^n\right)} \\
&= \frac{1}{N} \sum_{n=1}^{N} \left.\frac{\partial \ell\left(\hat{\mathbf{y}}, \mathbf{y}^n\right)}{\partial \hat{\mathbf{y}}} \frac{\partial \hat{\mathbf{y}}}{\partial \operatorname{vec}\left(\mathbf{W}_L\right)}\right|_{\hat{\mathbf{y}}=\sum_{i=0}^{L} \mathbf{W}_{L+1} \mathbf{W}_i \mathbf{v}_i\left(\mathbf{x}^n\right)} \\
&= \frac{1}{N} \sum_{n=1}^{N} \left.\frac{\partial \ell\left(\hat{\mathbf{y}}, \mathbf{y}^n\right)}{\partial \hat{\mathbf{y}}}\right|_{\hat{\mathbf{y}}=\mathbf{W}_{L+1} \sum_{i=0}^{L} \mathbf{W}_i \mathbf{v}_i\left(\mathbf{x}^n\right)} \left(\mathbf{v}_L\left(\mathbf{x}^n\right)^T \otimes \mathbf{W}_{L+1}\right) \\
&= \frac{1}{N} \sum_{n=1}^{N} \left(\left(\mathbf{v}_L\left(\mathbf{x}^n\right) \otimes \mathbf{W}_{L+1}^T\right) \underbrace{\left.\frac{\partial \ell\left(\hat{\mathbf{y}}, \mathbf{y}^n\right)}{\partial \hat{\mathbf{y}}}^T\right|_{\hat{\mathbf{y}}=\mathbf{W}_{L+1} \sum_{i=0}^{L} \mathbf{W}_i \mathbf{v}_i\left(\mathbf{x}^n\right)}}_{\mathbf{g}_r\left(\mathbf{x}^n\right)}\right)^T \\
&= \frac{1}{N} \sum_{n=1}^{N} \operatorname{vec}\left(\mathbf{W}_{L+1}^T \mathbf{g}_r\left(\mathbf{x}^n\right) \mathbf{v}_L\left(\mathbf{x}^n\right)^T\right)^T \\
&= \mathbf{0}.
\end{aligned} \tag{24}
$$

Equivalently, we can write the above first-order necessary condition into a matrix form

$$
\sum_{n=1}^{N} \mathbf{v}_L\left(\mathbf{x}^n\right) \mathbf{g}_r\left(\mathbf{x}^n\right)^T \mathbf{W}_{L+1} = \mathbf{0}. \tag{25}
$$

On the other hand, for the first-order necessary condition with respect to $\mathbf{W}_{L+1}$, we obtain

$$
\begin{aligned}
\frac{\partial \mathcal{R}}{\partial \operatorname{vec}\left(\mathbf{W}_{L+1}\right)} &= \frac{1}{N} \sum_{n=1}^{N} \left.\frac{\partial \ell\left(\hat{\mathbf{y}}, \mathbf{y}^n\right)}{\partial \operatorname{vec}\left(\mathbf{W}_{L+1}\right)}\right|_{\hat{\mathbf{y}}=\mathbf{W}_{L+1} \sum_{i=0}^{L} \mathbf{W}_i \mathbf{v}_i\left(\mathbf{x}^n\right)} \\
&= \frac{1}{N} \sum_{n=1}^{N} \left.\frac{\partial \ell\left(\hat{\mathbf{y}}, \mathbf{y}^n\right)}{\partial \hat{\mathbf{y}}} \frac{\partial \hat{\mathbf{y}}}{\partial \operatorname{vec}\left(\mathbf{W}_{L+1}\right)}\right|_{\hat{\mathbf{y}}=\sum_{i=0}^{L} \mathbf{W}_{L+1} \mathbf{W}_i \mathbf{v}_i\left(\mathbf{x}^n\right)} \\
&= \frac{1}{N} \sum_{n=1}^{N} \mathbf{g}_r\left(\mathbf{x}^n\right)^T \left(\left(\sum_{i=0}^{L} \mathbf{W}_i \mathbf{v}_i\left(\mathbf{x}^n\right)\right)^T \otimes \mathbf{I}_{N_o}\right) \\
&= \frac{1}{N} \sum_{n=1}^{N} \left(\left(\left(\sum_{i=0}^{L} \mathbf{W}_i \mathbf{v}_i\left(\mathbf{x}^n\right)\right) \otimes \mathbf{I}_{N_o}\right) \mathbf{g}_r\left(\mathbf{x}^n\right)\right)^T \\
&= \frac{1}{N} \sum_{n=1}^{N} \operatorname{vec}\left(\mathbf{g}_r\left(\mathbf{x}^n\right) \sum_{i=0}^{L} \mathbf{v}_i\left(\mathbf{x}^n\right)^T \mathbf{W}_i^T\right)^T \\
&= \mathbf{0}.
\end{aligned} \tag{26}
$$

The corresponding matrix form of the above condition is given by

$$
\sum_{i=0}^{L} \mathbf{W}_i \sum_{n=1}^{N} \mathbf{v}_i\left(\mathbf{x}^n\right) \mathbf{g}_r\left(\mathbf{x}^n\right)^T = \mathbf{0}. \tag{27}
$$

When $\mathbf{W}_{L+1}$ is full rank at a critical point, (25) implies $\sum_{n=1}^{N} \mathbf{v}_L (\mathbf{x}^n) \mathbf{g}_r (\mathbf{x}^n)^T = \mathbf{0}$ because the null space of $\mathbf{W}_{L+1}^T$ is degenerate according to Assumption 3. Then, applying such an implication to (27) along with Assumption 4, we obtain

$$\sum_{i=0}^{L-1} \mathbf{W}_i \sum_{n=1}^{N} \mathbf{v}_i (\mathbf{x}^n) \mathbf{g}_r (\mathbf{x}^n)^T = \mathbf{0} \implies \sum_{n=1}^{N} \mathbf{v}_i (\mathbf{x}^n) \mathbf{g}_r (\mathbf{x}^n)^T = \mathbf{0}, \quad i = 0, 1, \cdots, L-1. \quad (28)$$

Note that all the column vectors in $\begin{bmatrix} \mathbf{W}_0 & \mathbf{W}_1 & \cdots & \mathbf{W}_{L-1} \end{bmatrix}$ are linearly independent if and only if the linear inverse problem $\sum_{i=0}^{L-1} \mathbf{x}_i = \sum_{i=0}^{L-1} \mathbf{W}_i \mathbf{v}_i$ has a unique solution for $\mathbf{v}_0, \cdots, \mathbf{v}_{L-1}$. We have proved the statement (a).

On the other hand, when $\mathbf{W}_{L+1}$ is not full rank at a local minimizer, then there exists a perturbation on $\mathbf{W}_L$ such that the new point is still a local minimizer which has the same objective value. Let $(\mathbf{W}_L, \mathbf{W}_{L+1})$ be any local minimizer of $(\mathbf{P}_\phi)$ for which $\mathbf{W}_{L+1}$ is not full row rank. By the definition of a local minimizer, there exists some $\gamma > 0$ such that

$$\mathcal{R} (\mathbf{W}_L', \mathbf{W}_{L+1}'; \phi) \geq \mathcal{R} (\mathbf{W}_L, \mathbf{W}_{L+1}; \phi), \forall (\mathbf{W}_L', \mathbf{W}_{L+1}') \in B ((\mathbf{W}_L, \mathbf{W}_{L+1}), \gamma) \quad (29)$$

where $B$ is an open ball centered at $(\mathbf{W}_L, \mathbf{W}_{L+1})$ with the radius $\gamma$. Then $(\mathbf{W}_L + \mathbf{ab}^T, \mathbf{W}_{L+1})$ must also be a local minimizer for any nonzero $\mathbf{a} \in \mathcal{N}(\mathbf{W}_{L+1})$ and any sufficiently small nonzero $\mathbf{b} \in \mathbb{R}^{K_L}$ such that $(\mathbf{W}_L + \mathbf{ab}^T, \mathbf{W}_{L+1}) \in B ((\mathbf{W}_L, \mathbf{W}_{L+1}), \gamma/2)$. Substituting the minimizer $(\mathbf{W}_L + \mathbf{ab}^T, \mathbf{W}_{L+1})$ in (27) yields

$$\sum_{i=0}^{L-1} \mathbf{W}_i \sum_{n=1}^{N} \mathbf{v}_i (\mathbf{x}^n) \mathbf{g}_r (\mathbf{x}^n)^T + \left( \mathbf{W}_L + \mathbf{ab}^T \right) \sum_{n=1}^{N} \mathbf{v}_L (\mathbf{x}^n) \mathbf{g}_r (\mathbf{x}^n)^T = \mathbf{0}. \quad (30)$$

Subtracting (27) from the above equation, we obtain

$$\mathbf{ab}^T \sum_{n=1}^{N} \mathbf{v}_L (\mathbf{x}^n) \mathbf{g}_r (\mathbf{x}^n)^T = \mathbf{0}. \quad (31)$$

Multiplying both sides by $\mathbf{a}^T / \|\mathbf{a}\|_2^2$, we have

$$\mathbf{b}^T \sum_{n=1}^{N} \mathbf{v}_L (\mathbf{x}^n) \mathbf{g}_r (\mathbf{x}^n)^T = \mathbf{0} \implies \sum_{n=1}^{N} \mathbf{v}_L (\mathbf{x}^n) \mathbf{g}_r (\mathbf{x}^n)^T = \mathbf{0} \quad (32)$$

because $\mathbf{b} \neq \mathbf{0}$ can be arbitrary as long as it is sufficiently small. As a result, (28) is also true when $\mathbf{W}_{L+1}$ is not full row rank. We have proved the statement (b). $\qquad \square$

## A.4  Proof of Corollary 1

*Proof.* The proof of Theorem 1 has shown that every local minimizer $\left( \mathbf{W}_L^*, \mathbf{W}_{L+1}^* \right)$ of $(\mathbf{P}_\phi)$ establishes a corresponding global minimizer $\left( \mathbf{H}_0^*, \cdots, \mathbf{H}_L^* \right)$ in $(\mathbf{PA}_\phi)$ such that $\mathbf{H}_i^* = \mathbf{W}_{L+1}^* \mathbf{W}_i$ for $i = 0, 1, \cdots, L$. Therefore, it must be true that

$$\mathcal{R} \left( \mathbf{W}_{L_\alpha}^*, \mathbf{W}_{L_\alpha+1}^*, \boldsymbol{\alpha} \right) = \mathcal{A} \left( \mathbf{H}_0^*, \cdots, \mathbf{H}_{L_\alpha}^*, \mathbf{0}, \cdots, \mathbf{0}, \phi \right). \quad (33)$$

Next, by the convexity in Proposition 2, we have

$$\begin{aligned} \mathcal{R} \left( \mathbf{W}_{L_\alpha}^*, \mathbf{W}_{L_\alpha+1}^*, \boldsymbol{\alpha} \right) &= \mathcal{A} \left( \mathbf{H}_0^*, \cdots, \mathbf{H}_{L_\alpha-1}^*, \mathbf{H}_{L_\alpha}^*, \mathbf{0}, \cdots, \mathbf{0}, \phi \right) \\ &\overset{a}{\leq} \mathcal{A} \left( \mathbf{H}_0^*, \cdots, \mathbf{H}_{L_\beta}^*, \mathbf{0}, \cdots, \mathbf{0}, \phi \right) \\ &= \mathcal{R} \left( \mathbf{W}_{L_\beta}^*, \mathbf{W}_{L_\beta+1}^*, \boldsymbol{\beta} \right). \end{aligned} \quad (34)$$

The equality $a$ in (34) holds true by the relation $L_\beta < L_\alpha$. $\qquad \square$

## A.5 Proof of Corollary 2

*Proof.* By Theorem 1 (b),

$$\mathcal{R}\left(\mathbf{W}_0^*, \cdots, \mathbf{W}_{L+1}^*, \boldsymbol{\theta}_1^*, \cdots, \boldsymbol{\theta}_L^*\right) = \mathcal{R}\left(\mathbf{W}_L^*, \mathbf{W}_{L+1}^*, \boldsymbol{\phi}^*\right) = \mathcal{A}\left(\mathbf{H}_0^*, \cdots, \mathbf{H}_L^*, \boldsymbol{\phi}^*\right) \quad (35)$$

for any local minimizer $\left(\mathbf{H}_0^*, \cdots, \mathbf{H}_L^*\right)$ of (PA$_\phi$) using feature finding parameters $\boldsymbol{\phi}^*$. Then, by the convexity in Proposition 2, every local minimizer $\left(\mathbf{H}_0^*, \cdots, \mathbf{H}_L^*, \boldsymbol{\phi}^*\right)$ is a global minimzer of (PA$_\phi$) using $\boldsymbol{\phi}^*$. Hence, it must be true that

$$\begin{aligned}
\mathcal{R}\left(\mathbf{W}_0^*, \cdots, \mathbf{W}_{L+1}^*, \boldsymbol{\theta}_1^*, \cdots, \boldsymbol{\theta}_L^*\right) &= \mathcal{A}\left(\mathbf{H}_0^*, \cdots, \mathbf{H}_L^*, \boldsymbol{\phi}^*\right) \\
&\leq \mathcal{A}\left(\mathbf{H}_0^*, \mathbf{0}, \cdots, \mathbf{0}, \boldsymbol{\phi}^*\right) \\
&= \mathcal{A}\left(\mathbf{H}_0^*, \mathbf{0}, \cdots, \mathbf{0}, \boldsymbol{\psi}\right) \\
&= \min_{\mathbf{A} \in \mathbb{R}^{N_o \times N_{in}}} \frac{1}{N} \sum_{n=1}^{N} \ell\left(\mathbf{A}\mathbf{x}^n, \mathbf{y}^n\right)
\end{aligned} \quad (36)$$

for arbitrary $\boldsymbol{\psi}$ due to the zero prediction weights for $\mathbf{v}_1, \mathbf{v}_2, \cdots, \mathbf{v}_L$. We have proved the statement (a). If the inequality in (36) is strict, i.e.,

$$\mathcal{A}\left(\mathbf{H}_0^*, \cdots, \mathbf{H}_L^*, \boldsymbol{\phi}^*\right) < \mathcal{A}\left(\mathbf{H}_0^*, \mathbf{0}, \cdots, \mathbf{0}, \boldsymbol{\phi}^*\right), \quad (37)$$

then (36) implies

$$\mathcal{R}\left(\mathbf{W}_0^*, \cdots, \mathbf{W}_{L+1}^*, \boldsymbol{\theta}_1^*, \cdots, \boldsymbol{\theta}_L^*\right) < \min_{\mathbf{A} \in \mathbb{R}^{N_o \times N_{in}}} \frac{1}{N} \sum_{n=1}^{N} \ell\left(\mathbf{A}\mathbf{x}^n, \mathbf{y}^n\right). \quad (38)$$

We have proved the statement (b). $\qquad\square$

## A.6 Proof of Theorem 2

*Proof.* By Theorem 1 (a), every critical point with full rank $\mathbf{W}_{L+1}$ is a global minimizer of (P$_\phi$). Therefore, $\mathbf{W}_{L+1}$ must be rank-deficient at every saddle point. We have proved the statement (a).

We argue that the Hessian is neither positive semidefinite nor negative semidefinite at every saddle point. According to the proof of Proposition 1, there exists no point in the domain of the objective function of (P$_\phi$) at which the Hessian is negative semidefinite. If $\mathbf{W}_{L+1}$ is not full rank, then the positive semidefiniteness of the Hessian at every critical point becomes a sufficient condition for a local minimizer. This can be easily seen by replacing the convex loss with the squared loss in the proof for Theorem 1 and applying (18). We conclude that the Hessian must be indefinite at every saddle point under the assumptions; in other words, the Hessian has at least one strictly negative eigenvalue. We have proved the statement (b). $\qquad\square$

## A.7 Proof of Proposition 3

*Proof.* Note that (PD$_\phi$) is a convex optimization problem because its objective function is a nonnegative weighted sum of convex functions composited with affine mappings. Since (PD$_\phi$) is a convex optimization problem, it is true that

$$\mathcal{D}\left(\mathbf{W}_{L+1}^*; \boldsymbol{\phi}\right) \leq \min_{\mathbf{A} \in \mathbb{R}^{N_o \times N_{in}}} \mathcal{D}\left(\begin{bmatrix}\mathbf{A} & \mathbf{0} & \cdots & \mathbf{0}\end{bmatrix}; \boldsymbol{\phi}\right) = \min_{\mathbf{A} \in \mathbb{R}^{N_o \times N_{in}}} \frac{1}{N} \sum_{n=1}^{N} \ell\left(\mathbf{A}\mathbf{x}^n, \mathbf{y}^n\right) \quad (39)$$

for any local minimizer $\mathbf{W}_{L+1}^*$ of (PD$_\phi$) and arbitrary feature finding parameters $\boldsymbol{\phi}$. $\qquad\square$

## A.8 Proof of Proposition 4

*Proof.* Let $\mathbf{0}_{m \times n}$ be an $m$-by-$n$ zero matrix and $\mathbf{I}_{m \times n}$ be an $m$-by-$n$ matrix with ones on diagonal entries and zero elsewhere, i.e.,

$$[\mathbf{I}_{m \times n}]_{ij} = \begin{cases} 1, & i = j \\ 0, & i \neq j \end{cases}. \quad (40)$$

The superscript of every hyperparameter in this proof indicates the network type. We define $M^{\text{ResNEst}} = \sum_{i=0}^{L} K_i^{\text{ResNEst}} = M_L^{\text{DenseNEst}}$ and $K_i^{\text{ResNEst}} = D_i^{\text{DenseNEst}}$ for $i = 1, 2, \cdots, L$. Let

$$\boldsymbol{\Pi}_i = \begin{bmatrix} \mathbf{0}_{\left(\sum_{j=0}^{i-1} K_j^{\text{ResNEst}}\right) \times K_i^{\text{ResNEst}}} \\ \mathbf{I}_{K_i^{\text{ResNEst}} \times K_i^{\text{ResNEst}}} \\ \mathbf{0}_{\left(M^{\text{ResNEst}} - \sum_{j=0}^{i} K_j^{\text{ResNEst}}\right) \times K_i^{\text{ResNEst}}} \end{bmatrix} \tag{41}$$

for $i = 0, 1, \cdots, L$. Let $\mathbf{W}_{L+1}^{\text{ResNEst}} = \mathbf{W}_{L+1}^{\text{DenseNEst}}$ and $\mathbf{W}_i^{\text{ResNEst}} = \boldsymbol{\Pi}_i$ for $i = 0, 1, \cdots, L$. We define the function $\mathbf{G}_i$ in the ResNEst as

$$\mathbf{G}_i\left(\mathbf{x}_{i-1}\right) = \mathbf{Q}_i\left(\begin{bmatrix} \boldsymbol{\Pi}_0 & \boldsymbol{\Pi}_1 & \cdots & \boldsymbol{\Pi}_{i-1} \end{bmatrix}^T \mathbf{x}_{i-1}\right) \tag{42}$$

for $i = 1, \cdots, L$ where $\mathbf{x}_i \in \mathbb{R}^{M^{\text{ResNEst}}}$ is the residual representation in the ResNEst.

Based on such a construction, the feature finding weights $\phi$ in the ResNEst satisfies Assumption 4. Therefore, by Theorem 1 (b), the excess minimum empirical risk is zero or $\epsilon = 0$, i.e., the minimum value at every local minimizer of $(\mathrm{P}_\phi)$ is equivalent to the global minimum value in $(\mathrm{PA}_\phi)$. $\qquad\square$

## A.9 Important first- and second-order derivatives

We derive the Hessian in the proof of Proposition 1. Let $\mathbf{X} = \begin{bmatrix} \mathbf{x}^1 & \mathbf{x}^2 & \cdots & \mathbf{x}^N \end{bmatrix}$. Let $\hat{\mathbf{Y}}(\mathbf{X}) = \begin{bmatrix} \hat{\mathbf{y}}(\mathbf{x}^1) & \hat{\mathbf{y}}(\mathbf{x}^2) & \cdots & \hat{\mathbf{y}}(\mathbf{x}^N) \end{bmatrix}$ where each of column vectors is the ResNEst output given by the function $\hat{\mathbf{y}}(\mathbf{x}) = \mathbf{W}_{L+1} \sum_{i=0}^{L} \mathbf{W}_i \mathbf{v}_i(\mathbf{x})$. The empirical risk using the squared loss (up to a scaling factor) is defined as

$$\mathcal{R}\left(\mathbf{W}_L, \mathbf{W}_{L+1}; \phi\right) = \frac{1}{2}\frac{1}{N}\sum_{n=1}^{N}\left\|\hat{\mathbf{y}}\left(\mathbf{x}^n\right) - \mathbf{y}^n\right\|_2^2. \tag{43}$$

The Jacobian of $\mathcal{R}$ with respect to $\mathbf{W}_L$ is given by

$$\begin{aligned}
\frac{\partial \mathcal{R}}{\partial \mathrm{vec}(\mathbf{W}_L^T)} &= \frac{1}{2}\frac{1}{N}\frac{\partial}{\partial \mathrm{vec}(\mathbf{W}_L^T)}\sum_{n=1}^{N}\left\|\hat{\mathbf{y}}\left(\mathbf{x}^n\right) - \mathbf{y}^n\right\|_2^2 \\
&= \frac{1}{2}\frac{1}{N}\frac{\partial}{\partial \mathrm{vec}(\mathbf{W}_L^T)}\left\|\hat{\mathbf{Y}}(\mathbf{X}) - \mathbf{Y}\right\|_F^2 \\
&= \frac{1}{2}\frac{1}{N}\frac{\partial}{\partial \mathrm{vec}(\mathbf{W}_L^T)}\mathrm{vec}\left(\hat{\mathbf{Y}}(\mathbf{X})^T - \mathbf{Y}^T\right)^T \mathrm{vec}\left(\hat{\mathbf{Y}}(\mathbf{X})^T - \mathbf{Y}^T\right) \\
&= \frac{1}{N}\mathrm{vec}\left(\hat{\mathbf{Y}}(\mathbf{X})^T - \mathbf{Y}^T\right)^T \frac{\partial}{\partial \mathrm{vec}(\mathbf{W}_L^T)}\mathrm{vec}\left(\hat{\mathbf{Y}}(\mathbf{X})^T - \mathbf{Y}^T\right) \\
&= \frac{1}{N}\mathrm{vec}\left(\hat{\mathbf{Y}}(\mathbf{X})^T - \mathbf{Y}^T\right)^T \frac{\partial}{\partial \mathrm{vec}(\mathbf{W}_L^T)}\mathrm{vec}\left(\sum_{i=0}^{L}\mathbf{V}_i^T\mathbf{W}_i^T\mathbf{W}_{L+1}^T - \mathbf{Y}^T\right) \\
&= \frac{1}{N}\mathrm{vec}\left(\hat{\mathbf{Y}}(\mathbf{X})^T - \mathbf{Y}^T\right)^T \frac{\partial}{\partial \mathrm{vec}(\mathbf{W}_L^T)}\mathrm{vec}\left(\mathbf{V}_L^T\mathbf{W}_L^T\mathbf{W}_{L+1}^T\right) \\
&= \frac{1}{N}\mathrm{vec}\left(\hat{\mathbf{Y}}(\mathbf{X})^T - \mathbf{Y}^T\right)^T \frac{\partial}{\partial \mathrm{vec}(\mathbf{W}_L^T)}\left(\mathbf{W}_{L+1}\otimes\mathbf{V}_L^T\right)\mathrm{vec}\left(\mathbf{W}_L^T\right) \\
&= \frac{1}{N}\mathrm{vec}\left(\hat{\mathbf{Y}}(\mathbf{X})^T - \mathbf{Y}^T\right)^T \left(\mathbf{W}_{L+1}\otimes\mathbf{V}_L^T\right).
\end{aligned} \tag{44}$$

The Jacobian of $\mathcal{R}$ with respect to $\mathbf{W}_{L+1}$ is given by

$$
\begin{aligned}
\frac{\partial \mathcal{R}}{\partial \text{vec}(\mathbf{W}_{L+1}^T)} &= \frac{1}{2}\frac{1}{N}\frac{\partial}{\partial \text{vec}(\mathbf{W}_{L+1}^T)}\text{vec}\left(\hat{\mathbf{Y}}(\mathbf{X})^T - \mathbf{Y}^T\right)^T\text{vec}\left(\hat{\mathbf{Y}}(\mathbf{X})^T - \mathbf{Y}^T\right) \\
&= \mathbf{e}^T\frac{\partial}{\partial \text{vec}(\mathbf{W}_{L+1}^T)}\text{vec}\left(\hat{\mathbf{Y}}(\mathbf{X})^T - \mathbf{Y}^T\right) \\
&= \mathbf{e}^T\frac{\partial}{\partial \text{vec}(\mathbf{W}_{L+1}^T)}\text{vec}\left(\sum_{i=0}^{L}\mathbf{V}_i^T\mathbf{W}_i^T\mathbf{W}_{L+1}^T - \mathbf{Y}^T\right) \\
&= \mathbf{e}^T\frac{\partial}{\partial \text{vec}(\mathbf{W}_{L+1}^T)}\text{vec}\left(\sum_{i=0}^{L}\mathbf{V}_i^T\mathbf{W}_i^T\mathbf{W}_{L+1}^T\mathbf{I}_{N_o}\right) \\
&= \mathbf{e}^T\frac{\partial}{\partial \text{vec}(\mathbf{W}_{L+1}^T)}\left(\mathbf{I}_{N_o}\otimes\sum_{i=0}^{L}\mathbf{V}_i^T\mathbf{W}_i^T\right)\text{vec}\left(\mathbf{W}_{L+1}^T\right) \\
&= \mathbf{e}^T\left(\mathbf{I}_{N_o}\otimes\sum_{i=0}^{L}\mathbf{V}_i^T\mathbf{W}_i^T\right)
\end{aligned}
\tag{45}
$$

where we have used

$$
\mathbf{e} = \frac{1}{N}\text{vec}\left(\hat{\mathbf{Y}}(\mathbf{X})^T - \mathbf{Y}^T\right).
\tag{46}
$$

Now, we find each of the block matrices in the Hessian.

$$
\begin{aligned}
\frac{\partial^2\mathcal{R}}{\partial\text{vec}\left(\mathbf{W}_L^T\right)^2} &= \frac{\partial}{\partial\text{vec}\left(\mathbf{W}_L^T\right)}\left(\frac{\partial\mathcal{R}}{\partial\text{vec}(\mathbf{W}_L^T)}\right)^T \\
&= \frac{\partial}{\partial\text{vec}\left(\mathbf{W}_L^T\right)}\frac{1}{N}\left(\mathbf{W}_{L+1}^T\otimes\mathbf{V}_L\right)\text{vec}\left(\hat{\mathbf{Y}}(\mathbf{X})^T - \mathbf{Y}^T\right) \\
&= \frac{1}{N}\left(\mathbf{W}_{L+1}^T\otimes\mathbf{V}_L\right)\frac{\partial}{\partial\text{vec}\left(\mathbf{W}_L^T\right)}\text{vec}\left(\sum_{i=0}^{L}\mathbf{V}_i^T\mathbf{W}_i^T\mathbf{W}_{L+1}^T - \mathbf{Y}^T\right) \\
&= \frac{1}{N}\left(\mathbf{W}_{L+1}^T\otimes\mathbf{V}_L\right)\frac{\partial}{\partial\text{vec}\left(\mathbf{W}_L^T\right)}\text{vec}\left(\mathbf{V}_L^T\mathbf{W}_L^T\mathbf{W}_{L+1}^T\right) \\
&= \frac{1}{N}\left(\mathbf{W}_{L+1}^T\otimes\mathbf{V}_L\right)\frac{\partial}{\partial\text{vec}\left(\mathbf{W}_L^T\right)}\text{vec}\left(\mathbf{V}_L^T\mathbf{W}_L^T\mathbf{W}_{L+1}^T\right) \\
&= \frac{1}{N}\left(\mathbf{W}_{L+1}^T\otimes\mathbf{V}_L\right)\left(\mathbf{W}_{L+1}\otimes\mathbf{V}_L^T\right) \\
&= \frac{1}{N}\left(\mathbf{W}_{L+1}^T\mathbf{W}_{L+1}\otimes\mathbf{V}_L\mathbf{V}_L^T\right).
\end{aligned}
\tag{47}
$$

$$\frac{\partial^2 \mathcal{R}}{\partial \text{vec}\left(\mathbf{W}_{L+1}^T\right)^2} = \frac{\partial}{\partial \text{vec}\left(\mathbf{W}_{L+1}^T\right)} \left(\frac{\partial \mathcal{R}}{\partial \text{vec}\left(\mathbf{W}_{L+1}^T\right)}\right)^T$$

$$= \frac{1}{N} \frac{\partial}{\partial \text{vec}\left(\mathbf{W}_{L+1}^T\right)} \left(\mathbf{I}_{N_o} \otimes \sum_{i=0}^{L} \mathbf{W}_i \mathbf{V}_i\right) \text{vec}\left(\hat{\mathbf{Y}}(\mathbf{X})^T - \mathbf{Y}^T\right)$$

$$= \mathbf{F} \frac{\partial}{\partial \text{vec}\left(\mathbf{W}_{L+1}^T\right)} \text{vec}\left(\sum_{i=0}^{L} \mathbf{V}_i^T \mathbf{W}_i^T \mathbf{W}_{L+1}^T - \mathbf{Y}^T\right)$$

$$= \mathbf{F} \frac{\partial}{\partial \text{vec}\left(\mathbf{W}_{L+1}^T\right)} \text{vec}\left(\sum_{i=0}^{L} \mathbf{V}_i^T \mathbf{W}_i^T \mathbf{W}_{L+1}^T \mathbf{I}_{N_o}\right) \qquad (48)$$

$$= \mathbf{F} \left(\mathbf{I}_{N_o} \otimes \sum_{i=0}^{L} \mathbf{V}_i^T \mathbf{W}_i^T\right)$$

$$= \frac{1}{N} \left(\mathbf{I}_{N_o} \otimes \sum_{i=0}^{L} \mathbf{W}_i \mathbf{V}_i \left(\sum_{i=0}^{L} \mathbf{W}_i \mathbf{V}_i\right)^T\right)$$

where we have used

$$\mathbf{F} = \frac{1}{N} \left(\mathbf{I}_{N_o} \otimes \sum_{i=0}^{L} \mathbf{W}_i \mathbf{V}_i\right). \qquad (49)$$

$$\frac{\partial^2 \mathcal{R}}{\partial \text{vec}\left(\mathbf{W}_{L+1}^T\right) \partial \text{vec}\left(\mathbf{W}_L^T\right)}$$

$$= \frac{\partial}{\partial \text{vec}\left(\mathbf{W}_{L+1}^T\right)} \left(\frac{\partial \mathcal{R}}{\partial \text{vec}\left(\mathbf{W}_L^T\right)}\right)^T$$

$$= \frac{1}{N} \frac{\partial}{\partial \text{vec}\left(\mathbf{W}_{L+1}^T\right)} \left(\mathbf{W}_{L+1}^T \otimes \mathbf{V}_L\right) \text{vec}\left(\hat{\mathbf{Y}}(\mathbf{X})^T - \mathbf{Y}^T\right)$$

$$= \frac{1}{N} \left(\frac{\partial}{\partial \text{vec}\left(\mathbf{W}_{L+1}^T\right)} \left(\mathbf{W}_{L+1}^T \otimes \mathbf{V}_L\right)\right) \text{vec}\left(\hat{\mathbf{Y}}(\mathbf{X})^T - \mathbf{Y}^T\right) \qquad (50)$$

$$+ \frac{1}{N} \left(\mathbf{W}_{L+1}^T \otimes \mathbf{V}_L\right) \frac{\partial}{\partial \text{vec}\left(\mathbf{W}_{L+1}^T\right)} \text{vec}\left(\hat{\mathbf{Y}}(\mathbf{X})^T - \mathbf{Y}^T\right) \quad \text{(see (52))}$$

$$= \frac{1}{N} \left[\mathbf{I}_M \otimes \mathbf{V}_L \boldsymbol{\delta}_1 \quad \cdots \quad \mathbf{I}_M \otimes \mathbf{V}_L \boldsymbol{\delta}_{N_o}\right] + \frac{1}{N} \left(\mathbf{W}_{L+1}^T \otimes \mathbf{V}_L\right) \left(\mathbf{I}_{N_o} \otimes \sum_{i=0}^{L} \mathbf{V}_i^T \mathbf{W}_i^T\right)$$

$$= \frac{1}{N} \left[\mathbf{I}_M \otimes \mathbf{V}_L \boldsymbol{\delta}_1 \quad \cdots \quad \mathbf{I}_M \otimes \mathbf{V}_L \boldsymbol{\delta}_{N_o}\right] + \frac{1}{N} \left(\mathbf{W}_{L+1}^T \otimes \mathbf{V}_L \sum_{i=0}^{L} \mathbf{V}_i^T \mathbf{W}_i^T\right).$$

$$\frac{\partial^2 \mathcal{R}}{\partial \mathrm{vec}\left(\mathbf{W}_L^T\right) \partial \mathrm{vec}\left(\mathbf{W}_{L+1}^T\right)}$$

$$= \frac{\partial}{\partial \mathrm{vec}\left(\mathbf{W}_L^T\right)} \left(\frac{\partial \mathcal{R}}{\partial \mathrm{vec}\left(\mathbf{W}_{L+1}^T\right)}\right)^T$$

$$= \frac{1}{N} \frac{\partial}{\partial \mathrm{vec}\left(\mathbf{W}_L^T\right)} \left(\mathbf{I}_{N_o} \otimes \sum_{i=0}^{L} \mathbf{W}_i \mathbf{V}_i\right) \mathrm{vec}\left(\hat{\mathbf{Y}}(\mathbf{X})^T - \mathbf{Y}^T\right)$$

$$= \frac{1}{N} \left(\frac{\partial}{\partial \mathrm{vec}\left(\mathbf{W}_L^T\right)} \left(\mathbf{I}_{N_o} \otimes \sum_{i=0}^{L} \mathbf{W}_i \mathbf{V}_i\right)\right) \mathrm{vec}\left(\hat{\mathbf{Y}}(\mathbf{X})^T - \mathbf{Y}^T\right) \qquad (51)$$

$$+ \frac{1}{N} \left(\mathbf{I}_{N_o} \otimes \sum_{i=0}^{L} \mathbf{W}_i \mathbf{V}_i\right) \frac{\partial}{\partial \mathrm{vec}\left(\mathbf{W}_L^T\right)} \mathrm{vec}\left(\hat{\mathbf{Y}}(\mathbf{X})^T - \mathbf{Y}^T\right) \quad \text{(see (53) and (54))}$$

$$= \frac{1}{N} \begin{bmatrix} \mathbf{I}_M \otimes \boldsymbol{\delta}_1^T \mathbf{V}_L^T \\ \mathbf{I}_M \otimes \boldsymbol{\delta}_2^T \mathbf{V}_L^T \\ \vdots \\ \mathbf{I}_M \otimes \boldsymbol{\delta}_{N_o}^T \mathbf{V}_L^T \end{bmatrix} + \frac{1}{N} \mathbf{W}_{L+1} \otimes \sum_{i=0}^{L} \mathbf{W}_i \mathbf{V}_i \mathbf{V}_L^T.$$

Notice that we have used the following identities in (50) and (51).

$$\left(\frac{\partial}{\partial \mathrm{vec}\left(\mathbf{W}_{L+1}^T\right)} \left(\mathbf{W}_{L+1}^T \otimes \mathbf{V}_L\right)\right) \mathrm{vec}\left(\hat{\mathbf{Y}}(\mathbf{X})^T - \mathbf{Y}^T\right)$$

$$= \left(\frac{\partial}{\partial \mathrm{vec}\left(\mathbf{W}_{L+1}^T\right)} \left(\mathbf{W}_{L+1}^T \otimes \mathbf{V}_L\right)\right) \mathrm{vec}(\boldsymbol{\Delta})$$

$$= \sum_{j=1}^{N_o} \sum_{k=1}^{N} \left(\frac{\partial}{\partial \mathrm{vec}\left(\mathbf{W}_{L+1}^T\right)} \left(\left(\mathbf{W}_{L+1}^T\right)_j \otimes (\mathbf{V}_L)_k\right)\right) \delta_{k,j} \qquad (52)$$

$$= \left[\sum_{j=1}^{N_o} \sum_{k=1}^{N} \delta_{k,j} \frac{\partial \left(\mathbf{W}_{L+1}^T\right)_j \otimes (\mathbf{V}_L)_k}{\partial \left(\mathbf{W}_{L+1}^T\right)_1} \quad \cdots \quad \sum_{j=1}^{N_o} \sum_{k=1}^{N} \delta_{k,j} \frac{\partial \left(\mathbf{W}_{L+1}^T\right)_j \otimes (\mathbf{V}_L)_k}{\partial \left(\mathbf{W}_{L+1}^T\right)_{N_o}}\right]$$

$$= \left[\sum_{k=1}^{N} \delta_{k,1} \frac{\partial \mathrm{vec}\left((\mathbf{V}_L)_k \left(\mathbf{W}_{L+1}^T\right)_1^T\right)}{\partial \left(\mathbf{W}_{L+1}^T\right)_1} \quad \cdots \quad \sum_{k=1}^{N} \delta_{k,N_o} \frac{\partial \mathrm{vec}\left((\mathbf{V}_L)_k \left(\mathbf{W}_{L+1}^T\right)_{N_o}^T\right)}{\partial \left(\mathbf{W}_{L+1}^T\right)_{N_o}}\right]$$

$$= \left[\sum_{k=1}^{N} \delta_{k,1} \mathbf{I}_M \otimes (\mathbf{V}_L)_k \quad \cdots \quad \sum_{k=1}^{N} \delta_{k,N_o} \mathbf{I}_M \otimes (\mathbf{V}_L)_k\right]$$

$$= \left[\mathbf{I}_M \otimes \mathbf{V}_L \boldsymbol{\delta}_1 \quad \cdots \quad \mathbf{I}_M \otimes \mathbf{V}_L \boldsymbol{\delta}_{N_o}\right].$$

$$\left( \frac{\partial}{\partial \mathrm{vec}\left(\mathbf{W}_L^T\right)} \left( \mathbf{I}_{N_o} \otimes \sum_{i=0}^{L} \mathbf{W}_i \mathbf{V}_i \right) \right) \mathrm{vec}\left( \hat{\mathbf{Y}}(\mathbf{X})^T - \mathbf{Y}^T \right)$$

$$= \begin{bmatrix} \boldsymbol{\delta}_1^T \frac{\partial}{\partial \mathrm{vec}\left(\mathbf{W}_L^T\right)} \left( \sum_{i=0}^{L} \mathbf{V}_i^T \mathbf{W}_i^T \right)_1 \\ \vdots \\ \boldsymbol{\delta}_1^T \frac{\partial}{\partial \mathrm{vec}\left(\mathbf{W}_L^T\right)} \left( \sum_{i=0}^{L} \mathbf{V}_i^T \mathbf{W}_i^T \right)_M \\ \boldsymbol{\delta}_2^T \frac{\partial}{\partial \mathrm{vec}\left(\mathbf{W}_L^T\right)} \left( \sum_{i=0}^{L} \mathbf{V}_i^T \mathbf{W}_i^T \right)_1 \\ \vdots \\ \boldsymbol{\delta}_2^T \frac{\partial}{\partial \mathrm{vec}\left(\mathbf{W}_L^T\right)} \left( \sum_{i=0}^{L} \mathbf{V}_i^T \mathbf{W}_i^T \right)_M \\ \vdots \\ \boldsymbol{\delta}_{N_o}^T \frac{\partial}{\partial \mathrm{vec}\left(\mathbf{W}_L^T\right)} \left( \sum_{i=0}^{L} \mathbf{V}_i^T \mathbf{W}_i^T \right)_1 \\ \vdots \\ \boldsymbol{\delta}_{N_o}^T \frac{\partial}{\partial \mathrm{vec}\left(\mathbf{W}_L^T\right)} \left( \sum_{i=0}^{L} \mathbf{V}_i^T \mathbf{W}_i^T \right)_M \end{bmatrix}$$

$$= \begin{bmatrix} \boldsymbol{\delta}_1^T \mathbf{V}_L^T & \mathbf{0} & \mathbf{0} & \cdots & \mathbf{0} \\ \mathbf{0} & \boldsymbol{\delta}_1^T \mathbf{V}_L^T & \mathbf{0} & \cdots & \mathbf{0} \\ \mathbf{0} & \mathbf{0} & \boldsymbol{\delta}_1^T \mathbf{V}_L^T & \cdots & \mathbf{0} \\ \vdots & \vdots & \vdots & \ddots & \vdots \\ \mathbf{0} & \mathbf{0} & \mathbf{0} & \cdots & \boldsymbol{\delta}_1^T \mathbf{V}_L^T \\ \boldsymbol{\delta}_2^T \mathbf{V}_L^T & \mathbf{0} & \mathbf{0} & \cdots & \mathbf{0} \\ \mathbf{0} & \boldsymbol{\delta}_2^T \mathbf{V}_L^T & \mathbf{0} & \cdots & \mathbf{0} \\ \mathbf{0} & \mathbf{0} & \boldsymbol{\delta}_2^T \mathbf{V}_L^T & \cdots & \mathbf{0} \\ \vdots & \vdots & \vdots & \ddots & \vdots \\ \mathbf{0} & \mathbf{0} & \mathbf{0} & \cdots & \boldsymbol{\delta}_2^T \mathbf{V}_L^T \\ \vdots & \vdots & \vdots & \vdots & \vdots \\ \boldsymbol{\delta}_{N_o}^T \mathbf{V}_L^T & \mathbf{0} & \mathbf{0} & \cdots & \mathbf{0} \\ \vdots & \vdots & \vdots & \ddots & \vdots \\ \mathbf{0} & \mathbf{0} & \mathbf{0} & \cdots & \boldsymbol{\delta}_{N_o}^T \mathbf{V}_L^T \end{bmatrix} \tag{53}$$

$$= \begin{bmatrix} \mathbf{I}_M \otimes \boldsymbol{\delta}_1^T \mathbf{V}_L^T \\ \mathbf{I}_M \otimes \boldsymbol{\delta}_2^T \mathbf{V}_L^T \\ \vdots \\ \mathbf{I}_M \otimes \boldsymbol{\delta}_{N_o}^T \mathbf{V}_L^T \end{bmatrix}.$$

$$\left( \mathbf{I}_{N_o} \otimes \sum_{i=0}^{L} \mathbf{W}_i \mathbf{V}_i \right) \frac{\partial}{\partial \mathrm{vec}\left(\mathbf{W}_L^T\right)} \mathrm{vec}\left( \hat{\mathbf{Y}}(\mathbf{X})^T - \mathbf{Y}^T \right)$$

$$= \left( \mathbf{I}_{N_o} \otimes \sum_{i=0}^{L} \mathbf{W}_i \mathbf{V}_i \right) \left( \mathbf{W}_{L+1} \otimes \mathbf{V}_L^T \right) \tag{54}$$

$$= \mathbf{W}_{L+1} \otimes \sum_{i=0}^{L} \mathbf{W}_i \mathbf{V}_i \mathbf{V}_L^T.$$

# B Empirical results

In addition to the theoretical results, we also provide empirical results on image classification tasks to further understand our new principle-guided models. The goal of this empirical study is to answer the following question: *How do ResNEsts and A-ResNEsts perform compared to standard ResNets?*

## B.1 Datasets

The image classification tasks chosen in our empirical study are CIFAR-10 and CIFAR-100. The CIFAR-10 dataset [Krizhevsky, 2009] consists of 60000 $32 \times 32$ color images in 10 classes, with 6000 images per class. There are 50000 training images and 10000 test images. The CIFAR-100 dataset [Krizhevsky, 2009] is just like the CIFAR-10, except it has 100 classes containing 600 images each. The CIFAR-10 and CIFAR-100 datasets were collected by Alex Krizhevsky, Vinod Nair, and Geoffrey Hinton.

## B.2 Models and architectures

Every ResNEst was a standard ResNet without the batch normalization and Rectified Linear Unit (ReLU) at the final residual representation, i.e., their architectures are exactly the same before the final residual representation. Every BN-ResNEst was a standard ResNet without the ReLU at the final residual representation. In other words, a BN-ResNEst is a modified ResNEst because it adds a batch normalization layer at the final residual representation in the ResNEst. Such a modification can avoid gradient explosion during training and allow larger learning rates to be used. For A-ResNEsts, we applied 2-dimensional average pooling on each $\mathbf{v}_i$ going into each $\mathbf{H}_i$.

The standard ResNets used in this empirical study are wide ResNet 16-8 (WRN-16-8), WRN-40-4 [Zagoruyko and Komodakis, 2016], ResNet-110, and ResNet-20 [He et al., 2016b]. All these models use pre-activation residual blocks, i.e., they are in the pre-activation form [He et al., 2016b].

## B.3 Implementation details

The training procedure is exactly the same as the wide ResNet paper by Zagoruyko and Komodakis [2016]. The loss function was a cross-entropy loss. The batchsize was 128. All networks were trained for 200 epochs in total. The optimizer was stochastic gradient descent (SGD) with Nesterov momentum. The momentum was set to 0.9. The weight decay was 0.0005. The learning rate was initially set to 0.1 (0.01 for ResNEsts to avoid gradient explosion) and decreased by a factor of 5 after training 60, 120, and 160 epochs. Learning rates 0.1 and 0.05 both led to gradient explosion in training ResNEsts, so we used 0.01 for ResNEsts to avoid divergence. A-ResNEsts and BN-ResNEsts do not have such an issue.

In addition, we followed the same moderate data augmentation and preprocessing techniques in the wide ResNet paper by Zagoruyko and Komodakis [2016]. For the moderate data augmentation, a random horizontal flip and a random crop from a image padded by 4 pixels on each side are applied on the training set. For preprocessing, standardization is applied to every image including the training set and the test set. The mean and the standard deviation are computed from the training set.

Code is available at https://github.com/kjason/ResNEst.

## B.4 Comparison

Our empirical results are summarized in the two tables below in terms of classification accuracy and number of parameters. The classification accuracy is an average of 7 trials with different initializations. The number of parameters is shown in the unit of million. "1.0M" means one million parameters.

Table 1: CIFAR-10.

| Model Archit. | Standard | ResNEst | BN-ResNEst | A-ResNEst |
|---|---|---|---|---|
| WRN-16-8 | 95.56% (11M) | 94.39% (11M) | 95.48% (11M) | 95.29% (8.7M) |
| WRN-40-4 | 95.45% (9.0M) | 94.58% (9.0M) | 95.61% (9.0M) | 95.48% (8.4M) |
| ResNet-110 | 94.46% (1.7M) | 92.77% (1.7M) | 94.52% (1.7M) | 93.97% (1.7M) |
| ResNet-20 | 92.60% (0.27M) | 91.02% (0.27M) | 92.56% (0.27M) | 92.47% (0.24M) |

Table 2: CIFAR-100.

| Model Archit. | Standard | ResNEst | BN-ResNEst | A-ResNEst |
|---|---|---|---|---|
| WRN-16-8 | 79.14% (11M) | 75.43% (11M) | 78.99% (11M) | 78.74% (8.9M) |
| WRN-40-4 | 79.08% (9.0M) | 75.16% (9.0M) | 78.97% (9.0M) | 78.62% (8.7M) |
| ResNet-110 | 74.08% (1.7M) | 69.08% (1.7M) | 73.95% (1.7M) | 72.53% (1.9M) |
| ResNet-20 | 68.56% (0.28M) | 64.73% (0.28M) | 68.47% (0.28M) | 68.16% (0.27M) |

## B.5 A-ResNEsts empirically exhibit competitive performance to standard ResNets

Empirical results in Section B.4 show that A-ResNEsts in general exhibit competitive classification accuracy with fewer parameters compared to standard ResNets; and ResNEsts are not as good as A-ResNEsts. The A-ResNEts in most cases have fewer parameters than the ResNEsts and standard ResNets because they do not have the layers $\mathbf{W}_L$ and $\mathbf{W}_{L+1}$; and the number of prediction weights in $\mathbf{H}_0, \mathbf{H}_1, \cdots, \mathbf{H}_L$ is usually not larger than the number of weights in $\mathbf{W}_L$ and $\mathbf{W}_{L+1}$ (see Figure 1 and Figure 2). Note that A-ResNEsts can have more parameters than standard ResNets when the depth and the output dimension are very large, e.g., the A-ResNEst model under the architecture ResNet-110 for CIFAR-100 in Table 2.

## B.6 A BN-ResNEst slightly outperforms a standard ResNet when the network is very deep on the CIFAR-10 dataset

Empirical results in Table 1 show that BN-ResNEsts slightly outperform standard ResNets and A-ResNEsts in the architectures WRN-40-4 and ResNet-110 on the CIFAR-10 dataset. For architectures WRN-16-8 and ResNet-20, BN-ResNEsts remain competitive performance compared to standard ResNets. Notice that WRN-40-4 and ResNet-110 are much deeper than WRN-16-8 and ResNet-20. Therefore, these empirical results suggest that keeping the batch normalization and simply dropping the ReLU at the final residual representation in standard pre-activation ResNets can improve the test accuracy on CIFAR-10 when the network is very deep. However, if the batch normalization at the final residual representation is also dropped, then the test accuracy is noticeably lower.