# OpenReview forum: "ResNEsts and DenseNEsts: Block-based DNN Models with Improved Representation Guarantees"
_NeurIPS.cc/2021/Conference — NeurIPS 2021 Poster_

### Official Review · Reviewer_Mr4e · 2021-07-14

**Rating:** 6
**Confidence:** 2

**Summary:**

The authors present several theoretical properties of a ResNet variant that is similar to a standard ResNet only without one of the non-linearities. Variants of DenseNets are also suggested with improved guarantees. Such variants are much closer to the real ResNet that is used in computer vision applications than the models that were previously analyzed.

**Ethical Concerns:**

No concerns.

**Limitations And Societal Impact:**

The authors did not address those, but I do not see any issues or have any suggestions for improvement.

**Main Review:**

The paper is interesting and introduces some important theoretical concepts and proofs. However, it is tough to follow and understand. Even starting at Section 2.1, I could not quite understand for sure what the ResNEst model is. This has to be improved somehow.

Can the authors show some results regarding how ResNEst performs in simple classification tasks compared to a standard ResNet? Usually, simpler variants with guarantees perform worse than the original, and it would be interesting to see just how much ResNEst loses in classification accuracy to get these guarantees.

**Time Spent Reviewing:**

4

---

> ### Author Response · Authors · 2021-08-10
> **Experimental results and clarification on the definition of ResNEsts**
>
> Thank you very much for your comments that will most certainly help us improve the paper, given the opportunity. We will make necessary modifications to the main text and add empirical results in the supplementary material as we revise the manuscript.
>
> ### Below we respond to the reviewer's specific concern and suggestion:
> >The paper is interesting and introduces some important theoretical concepts and proofs. However, it is tough to follow and understand. Even starting at Section 2.1, I could not quite understand for sure what the ResNEst model is. This has to be improved somehow.
>
> The ResNEst model is shown in Figure 2 and defined in lines 124 to 133. The only difference between the ResNEst and the standard ResNet is the nonlinearity at the last residual representation (Figure 2 also shows such a difference). We are sorry about your experience with the reading of the paper. We will work on improving our main text and making it easy to read.
> >Can the authors show some results regarding how ResNEst performs in simple classification tasks compared to a standard ResNet? Usually, simpler variants with guarantees perform worse than the original, and it would be interesting to see just how much ResNEst loses in classification accuracy to get these guarantees.
>
> We thank the reviewer for this important suggestion. We strongly agree with the reviewer that empirical results for our new models would be helpful to support the derived theoretical guarantees.
>
> We evaluated ResNEsts, A-ResNEsts, and standard ResNets on CIFAR-10 and CIFAR-100 for 4 different architectures (different depth and width, they are WRN-40-4, WRN-16-8, ResNet-110, and ResNet-20). Please see our experimental results in the post above.
>
> ---
>
> Based on the significant contributions of the paper, we feel that the paper is deserving of a higher rating. We sincerely hope the reviewer will reconsider based on the responses to the issues raised and the experimental evidence provided.

---

> > ### Comment · Reviewer_Mr4e · 2021-08-24
> > **Response**
> >
> > Thank you for the response and the new results. I have upgraded my score for this paper to 6.

---

### Official Review · Reviewer_SQ4N · 2021-07-16

**Rating:** 8
**Confidence:** 2

**Summary:**

This paper defines slightly modified variants of commonly used ResNet and DenseNet architectures, and shows that these architectures have attractive theoretical properties - increasing depth does not decrease performance (which in particular means that they should be guaranteed to outperform linear models in terms of training loss).

**Limitations And Societal Impact:**

There is very little discussion of these. However, I do not think any negative societal impact would apply as this paper simply advances theoretical understanding.

**Main Review:**

The problem addressed (generalization properties of realistic modern neural network architectures) is important and relevant to the machine learning community, and the solution is novel and represents a substantial improvement over existing work. The paper is fairly well-written and clear.

Although this is a theory paper, it would benefit from experiments to compare standard ResNets with ResNEsts and especially A-ResNEsts (and similarly DenseNets and DenseNEsts), since these are new model architectures being proposed. Although the theoretical justification for these new architectures appears to be sound, showing that they also work well in practice would substantially improve the paper, as that combined with the theoretical guarantees would provide strong support for practitioners to actually use these architectures. Although ResNEsts are a trivial modification of standard ResNets, this is not as true for the other new architectures proposed (and even ablating the effect of the minor difference between ResNet and ResNEst would be very useful).

**Time Spent Reviewing:**

3

---

> ### Author Response · Authors · 2021-08-10
> **Experimental results**
>
> Thank you very much for your comments that will most certainly help us improve the paper, given the opportunity. We will add empirical results in the supplementary material as we revise the manuscript.
>
> ### Below we respond to the reviewer’s suggestion:
> >Although this is a theory paper, it would benefit from experiments to compare standard ResNets with ResNEsts and especially A-ResNEsts (and similarly DenseNets and DenseNEsts), since these are new model architectures being proposed. Although the theoretical justification for these new architectures appears to be sound, showing that they also work well in practice would substantially improve the paper, as that combined with the theoretical guarantees would provide strong support for practitioners to actually use these architectures. Although ResNEsts are a trivial modification of standard ResNets, this is not as true for the other new architectures proposed (and even ablating the effect of the minor difference between ResNet and ResNEst would be very useful).
>
> We thank the reviewer for this important suggestion. We strongly agree with the reviewer that empirical results for our new models would bring a much broader impact, given the derived theoretical guarantees.
>
> We evaluated ResNEsts, A-ResNEsts, and standard ResNets on CIFAR-10 and CIFAR-100 for 4 different architectures (different depth and width, they are WRN-40-4, WRN-16-8, ResNet-110, and ResNet-20). Please see our experimental results in the post above.

---

> > ### Comment · Reviewer_SQ4N · 2021-09-01
> > **Thank you for the detailed experimental evaluation**
> >
> > Under the assumption that these evaluations will be described and included in the final paper, my primary criticism of the paper has been addressed. Therefore, I raise my score to 8.
> >
> > Minor comment: In Figure 1, the sentence "The A-ResNEst is always better than the ResNEst" should be clarified to say that this is referring to training loss (as the precise meaning of "better" has not been fully explained at this point in the paper).

---

> > > ### Author Response · Authors · 2021-09-02
> > > **Experimental results will be included and we will improve the sentence in Figure 1**
> > >
> > > We thank the reviewer for the positive feedback. The experiments and their results will be described and included in the supplementary material (Appendix). In the main text, we will point out these empirical findings and refer the reader to the Appendix.
> > >
> > > We thank the reviewer for the additional comment. We will point out such a superiority belongs to training and improve the sentence.

---

> ### Author Response · Authors · 2021-08-30
> **We would be happy to answer any further questions**
>
> Once again, thank you for your positive review. It was encouraging. We have provided experimental results and they are quite promising. We hope you find them also interesting; and, reinforce your positive view of the paper and make you more supportive of our work.

---

### Official Review · Reviewer_zqDC · 2021-07-16

**Rating:** 7
**Confidence:** 4

**Summary:**

This paper studies a variant of ResNet/DenseNet, which removes the final nonlinearity. This makes the output a linear combination of a set of basis functions, which can be viewed as features extracted by the various blocks of the architecture. Based on this formulation, results regarding the optimization minima are examined. An augmented version of the architecture in question is also analyzed.

**Limitations And Societal Impact:**

The authors have adequately described the limitations present in each theorem/proposition made. Depending on how the $v_i(x)$ are obtained, I believe it is worthwhile to discuss this as well.

Regarding negative social impacts, this is a purely theoretical work, and the models presented do not have any issues of such kind to be discussed.

**Main Review:**

The paper analyzes ResNEsts, which are essentially ResNets without a final nonlinearity. This characterization leads to some interesting results:
- Assuming all layers except the last one are fixed, the output can be viewed due to this formulation as a linear combination of basis functions $v_i(x)$, each outputted by the previous blocks of the network.
- This characterization also serves as an inspiration for a slightly different architecture, Augmented ResNEsts. In those, the output of each residual block is kept and used at the end of the model, multiplied by another weight matrix $H_i$.
The paper also defines DenseNEsts, as an extension of the previous architecture similar to DenseNets, and also proves that they can be seen as a sufficiently wide ResNEst.

This work serves as a theoretical analysis of the above variation of ResNets, and the theorems/propositions included are technically sound. Proofs for some training properties (namely, directions of negative curvature in saddle points, and that wide ResNEsts attain their proposed ERLBs) are also included. Nevertheless, there are some issues which limit the impact of the paper:
- Regarding ResNEsts, all theorems are done under the assumption that all weight matrices except the two last ones are fixed. Note though that this is done to keep the functions $v_i(x)$ fixed, and if I understand correctly the intent of the authors was to show via Theorem 1 that stacking extra blocks in the wide ResNEst will not decrease performance (since the A-ResNEst essentially includes ResNEsts of all previous block sizes). In this case, it may be less of a limit of novelty and more of a clarity issue.
- Similarly, regarding A-ResNEsts, due to keeping the weight matrices of the intermediate layers (and thus, the $v_i(x)$) fixed, the analysis presented in this paper is the same as the analysis that would be done for a linear predictor, which has as input all the features $v_i(x)$ and a block diagonal weight matrix, with each part being equal to $H_i$. This is a very specific case of the proposed architecture.
- Corollary 1 appears to be self-evident (as a class of models, ResNEsts already include linear predictors).

This submission is clear in general and mostly easy to follow, although I do have some questions for the authors:
- If I am not mistaken, Remark 2 seems to imply that the intended use of ResNEsts is that training is done layer-by-layer, and the result is that doing so always improves the performance of the output. If this is indeed the case, then I suggest including it more prominently in the paper.
- In lines 252-253, a condition is stated in order for Assumption 4 to be satisfied. Does this mean that Assumptions 3 and 4 can be combined?
The clarity of the text could also be improved by explicitly defining optimization problems (so that they can easily be referred to later), instead of keeping them inside the text.

In general, the paper has several different contributions, but most of them appear limited in scope and application. I think it would be useful for the authors to highlight how the basis functions $v_i(x)$ are obtained, or if they are assumed to be given for the application of the ResNEst.

**Post-rebuttal comment**:

After the author response, I have updated my score, based on my comments below.

**Time Spent Reviewing:**

8

---

> ### Author Response · Authors · 2021-08-10
> **Clarification on theoretical results**
>
> Thank you very much for your comments that will most certainly help us improve the paper, given the opportunity. We will make necessary modifications to the main text as we revise the manuscript.
>
> ### Below we address specific concerns:
> >Assuming all layers except the last one are fixed, the output can be viewed due to this formulation as a linear combination of basis functions $v_i(x)$, each outputted by the previous blocks of the network.
>
> In hindsight, this may look simple but such a simple characterization was missing to date and its importance and impact are significant. This viewpoint has enabled many of the interesting results in the paper.
> >Regarding ResNEsts, all theorems are done under the assumption that all weight matrices except the two last ones are fixed. Note though that this is done to keep the functions $v_i(x)$ fixed, and if I understand correctly the intent of the authors was to show via Theorem 1 that stacking extra blocks in the wide ResNEst will not decrease performance (since the A-ResNEst essentially includes ResNEsts of all previous block sizes). In this case, it may be less of a limit of novelty and more of a clarity issue.
>
> This is a very good question. The original intention of Theorem 1 is not about layerwise or blockwise training, but we are glad that the reviewer raises such an interpretation for Theorem 1. We will clarify Theorem 1 with two different interpretations as we revise the manuscript. Notice the statement in Theorem 1, “...then in ($P_{\\phi}$) under any $\\phi$ such that Assumption 4 holds…,” so the result is more generalized than the reviewer’s interpretation. We will highlight the “under any $\\phi$” in our Theorem 1.
>
> Notice that in Theorem 1, we did not use $(P)$ and then characterize all its local minimizers that satisfy Assumption 4. Instead, we chose to characterize all local minimizers of ($P_{\\phi}$) under any $\\phi$ that satisfy Assumption 4 due to generality. The reason can be found in Remark 1. **Hence, Theorem 1 and Theorem 2 are not limited to fixing any weights during training; that is, they apply to both normal training and blockwise training procedures**.
>
> Although the reviewer didn’t point this out, we guess the reviewer’s concern in fixing weight matrices should apply to ERLBs established by A-ResNEsts. When two local minimizers of ($P$) that satisfy Assumption 4 have different feature finding parameters $\\phi$, they may attain different guaranteed lower bounds. However, Corollary 1 guarantees that they are at least better than any linear predictors.
> >Similarly, regarding A-ResNEsts, due to keeping the weight matrices of the intermediate layers (and thus, the $v_i(x)$) fixed, the analysis presented in this paper is the same as the analysis that would be done for a linear predictor, which has as input all the features $v_i(x)$ and a block diagonal weight matrix, with each part being equal to $H_i$. This is a very specific case of the proposed architecture.
>
> Fixing feature finding weights in A-ResNEsts is only necessary when A-ResNEsts serve as lower bounds for ResNEsts. Using Remark 1 (the same logic above), Proposition 2 guarantees that any local minimizer of ($P$) is at most equally good as any local minimizer of ($PA_{\\phi}$) which may or may not be a local minimizer of ($PA$). In such a sense, an A-ResNEst is better than a ResNEst. If they obtain two local minimizers with different feature finding weights, nothing can be concluded as we do not make assumptions about the residual functions.
> >Corollary 1 appears to be self-evident (as a class of models, ResNEsts already include linear predictors).
>
> Corollary 1 may be intuitive, but theoretically proving every local minimum is always better than any linear predictors is nontrivial and an active research topic [Shamir, 2018, Kawaguchi and Bengio, 2019, Yun et al., 2019]. To the best of our knowledge, Corollary 1 is the first theoretical guarantee for vector-valued ResNet-like models that have arbitrary residual blocks to outperform any linear predictors. In addition, Corollary 1 is a generalization of the previous results [Shamir, 2018, Kawaguchi and Bengio, 2019, Yun et al., 2019]. Corollary 1 advances this line of theoretical works. The impact of this result can be quite significant in fields like signal processing, wireless communications and others. The guarantees and superiority over linear estimators in spite of using learned nonlinear features open up new possibilities for applications in these fields that heavily rely on simple linear predictors. For example, many linear systems in communication, medical imaging, or feedback/echo cancellation may be replaced by ResNEsts or A-ResNEsts without any concern for harming performance.
> >If I am not mistaken, Remark 2 seems to imply that the intended use of ResNEsts is that training is done layer-by-layer, and the result is that doing so always improves the performance of the output. If this is indeed the case, then I suggest including it more prominently in the paper.
>
> Our intention in this paper has been to train all the weights in a network as a whole. Our theoretical results apply to both normal training and blockwise training procedures. This interpretation for Remark 2 from the reviewer is similar to the previous issue in Theorem 1. Again, we can have two different interpretations. Besides the reviewer’s interpretation, Remark 2 guarantees that at any local minimizer of ($P$) satisfying assumptions, the minimum empirical risk obtained by the best linear model using $\\mathbf{x}_i$ as the input representation is lower than or equal to the one using $\\mathbf{x}_j$ for $i > j$. We do not require any blockwise or layerwise training to establish Remark 2.
>
> In short, once a wide ResNEst with bottleneck blocks has been trained, any $\\mathbf{x}_i$ is better than $\\mathbf{x}_j$ for $i > j$ in the representational sense.
> >In lines 252-253, a condition is stated in order for Assumption 4 to be satisfied. Does this mean that Assumptions 3 and 4 can be combined?
>
> Assumption 3 and 4 are both required in order to prove Theorem 1 (see lines 545 to 548) because the main idea of the proof is to show that every local minimizer of ($P_{\\phi}$) has a corresponding global minimizer in ($PA_{\\phi}$) such that $H_i^*=W_{L+1}^*W_i$ (see lines 538 to 540).
>
> The bottleneck condition is an implicit assumption because any solution of a ResNEst that satisfies Assumption 4 must also satisfy the bottleneck condition. Notice that Assumption 4 does not involve $W_L$ and $W_{L+1}$ so it has nothing to do with the output dimension $N_O$.
> >The clarity of the text could also be improved by explicitly defining optimization problems (so that they can easily be referred to later), instead of keeping them inside the text.
>
> We are sorry about your experience with the reading of the paper. We will work on improving our optimization definitions and write down whole optimization problems in a dedicated space. We put them in the text because empirical risk minimization is generally known and the space is constrained.
> >In general, the paper has several different contributions, but most of them appear limited in scope and application.
>
> To the best of our knowledge, Theorem 1 generalizes and advances the line of recent seminal theoretical works on ResNets [Shamir, 2018, Kawaguchi and Bengio, 2019, Yun et al., 2019]. We hope our responses above have addressed the reviewer’s concerns regarding our theoretical results.
> >I think it would be useful for the authors to highlight how the basis functions $v_i(x)$ are obtained, or if they are assumed to be given for the application of the ResNEst.
>
> The weights for these basis functions (feature finding weights) are learned from normal training with data. To be more specific, feature finding weights are found by training a network, i.e., solving ($P$) or ($PA$). They are part of a local minimizer of the empirical risk minimization problem on the whole weights of a network.
>
> ---
>
> Based on the significant contributions of the paper, we feel that the paper is deserving of a higher rating. We sincerely hope the reviewer will reconsider based on the responses to the issues raised and the experimental evidence provided.

---

> > ### Comment · Reviewer_zqDC · 2021-08-25
> > **Post-Rebuttal Update**
> >
> > Thank you very much for your detailed responses to both my and the rest of the reviewers' comments.
> >
> > Regarding Corollary 1, I see now that it is more general than I originally stated, given that it applies to all local minima of ResNEsts, which is not as straightforward as I previously mentioned. Moreover, I thank you for clarifying whether or not $\mathbf{\phi}$ is assumed to be fixed.
> >
> > In light of the above, I have raised my original score. I believe that it would be of great benefit for the clarity of the paper if the importance of Remark 1 on how the results for $(P_{\mathbf{\phi}})$ can be transferred to $(P)$ was further highlighted in the interpretation of the main theorems (as well as possibly in their respective proofs in the appendix).

---

> > > ### Author Response · Authors · 2021-08-26
> > > **We will further highlight Remark 1 for clarity**
> > >
> > > We thank the reviewer for the update. In addition to the improvements we mentioned in the first response to the reviewer, we will also further highlight the importance of Remark 1 as we revise the manuscript. In particular, we will add a proof for Remark 1, and further highlight its usefulness for main theorems.

---

### Official Review · Reviewer_AWNq · 2021-07-17

**Rating:** 5
**Confidence:** 3

**Summary:**

The authors proposed a couple modification of ResNet in order to improve the loss landscape. The main idea behind the modifications, if I understand correctly,  is to guarantee a direct skip connection (dense matrix or product of dense matrix) from the hidden features to the logits. The authors provide a  couple theoretical claims regarding the proposed models. No experimental results are provided.

**Limitations And Societal Impact:**

Theoretical understanding of ResNet is important and indeed very important! Principle-guided approach to design architecture is also equally important. However, I don't think the paper made significant advances in these two problems, in particular, without strong empirical support. I would encourage the authors to provide experimental results to demonstrate the superior of the proposed architectures.

**Main Review:**

Overall, I found the presentation of the paper unsatisfying and difficult to follow. From the abstract and introduction, I find myself difficult to extract the main theme and main contribution of the paper. Only after going thorough the papers twice, I finally realized the paper is about landscape analysis of ResNets and its variants. Regardless of the results of the paper, presentation requires substantial improvement and empirical supports are needed to support major claims. Moreover, methods/modifications that improve loss landscape of the SOTA models (e.g. ResNet) more often than not may not be useful at all, unless they could improve the original performance, or on par with it and improves the optimization, or simplifies the models and etc. As such, I do not support acceptance.

In what follows, I try my best to summarize the main contribution and ideas of the paper.
The main complaint about ResNet of the paper is the *non-linearity* layer before the linear classifier and the authors propose to remove it. The main benefit of doing so is the logits $y$ could be written as a linear combination of hidden features of the networks, namely,
$$
y(x) = \sum_{i=1}^L A_i v_i(x)
$$
where $v_i(x)$ is the hidden feature and $A_i$ is an affine transform which is product of two matrices. Under some rank assumptions, the authors claim that loss function associated to this modified model is non-convex but enjoys a good property: every critical point that is not a local minimum is a saddle point (proposition 1). Obvious, the fact $A_i$ being a product of two dense matrices creates many saddle points and the author propose add a skip-connection from $v_i$ directly to the logits layer and $A_i$ could be replaced by a dense matrix $H_i$, i.e.
$$
y(x) = \sum_{i=1}^L H_i v_i(x)
$$
The authors proceed to conclude that the new optimization problem is convex, which seems obvious since the new model is a linear function function the hidden features. They then apply similar tricks to DenseNet.

Overall, these landscape results seem not very interesting nor significancy, at least to me.

**Time Spent Reviewing:**

4

---

> ### Author Response · Authors · 2021-08-09
> **Experimental results and clarification on our models and theoretical results**
>
> Thank you very much for your comments that will most certainly help us improve the paper, given the opportunity. We will make necessary modifications to the main text and add empirical results in the supplementary material as we revise the manuscript.
>
> ### Below we address specific concerns and misunderstandings:
>
> >The authors proposed a couple modification of ResNet in order to improve the loss landscape.
>
> The main purpose of our models and results are actually not about improving the loss landscape of a ResNet, while the analyses in our paper could potentially benefit this objective too. Indeed, the focus of this paper is to take a step towards understanding how ResNets work and, more importantly, why learning better ResNets is as easy as stacking more blocks via developing theoretical guarantees on local minimizers.
> >The main idea behind the modifications, if I understand correctly, is to guarantee a direct skip connection (dense matrix or product of dense matrix) from the hidden features to the logits.
>
> Actually, the main idea behind such a modification is to make the model as close to standard ResNets as possible, while circumventing mathematical intractabilities in ResNets, so that we are able to derive theoretical guarantees for local minimizers without putting assumptions on residual functions or data. Indeed, many previous related works also dropped the nonlinearity at the last residual representation for tractability; for example see [Hardt and Ma, 2016, Shamir, 2018, Kawaguchi and Bengio, 2019, Yun et al., 2019]. However, they required more assumptions and the models considered are even far away from the standard ResNets. Following this line of research, the results presented in our paper are a major step forward in understanding ResNets as we only impose minimum modification (dropping the last nonlinearity) of ResNet models.
>
> More importantly, such a modification reveals a linear relationship between the output of the ResNEst and the input linear feature and the nonlinear feature in each block. In hindsight, this may look simple but such a simple characterization was missing to date and its importance and impact are significant. This viewpoint has enabled many of the interesting results in the paper.
>
> In fields like signal processing, wireless communications, and others, this paper can have significant impacts.  The guarantees and superiority over linear estimators in spite of using learned nonlinear features open up new possibilities for applications in these fields that heavily rely on simple linear predictors. For example, many linear systems in communication, medical imaging, or feedback/echo cancellation may be replaced by ResNEsts or A-ResNEsts without any concern for harming performance.
> >No experimental results are provided.
>
> Although this is a theoretical paper, we will add our empirical results in the supplementary material as we revise the manuscript. Please see our experimental results in the post above.
> >Overall, I found the presentation of the paper unsatisfying and difficult to follow. From the abstract and introduction, I find myself difficult to extract the main theme and main contribution of the paper.
>
> We are sorry about your experience with the reading of the paper. We will work on improving our main text and making it easy to read. Particularly, we will highlight that we derive and prove the first theoretical guarantee for generic vector-valued L-block ResNet-like models (Theorem 1). The significance of Theorem 1 is stated in section 3 of our paper.
> >Only after going thorough the papers twice, I finally realized the paper is about landscape analysis of ResNets and its variants.
>
> This paper actually does not analyze the landscape of a ResNet (or standard ResNet). The ResNets are in general analytically intractable. Therefore, we construct close approximations: ResNEsts and A-ResNEsts in order to derive mathematical guarantees. We find most theoretical papers do not highlight the difference (the nonlinearity at the last representation), but we do. Our ResNEst models are closer to standard ResNets compared to the line of works of [Shamir, 2018, Kawaguchi and Bengio, 2019, Yun et al., 2019].
>
> >empirical supports are needed to support major claims
>
> If we understand correctly, the major claims here refer to our theorems, corollaries, and propositions. All of their proofs are provided in Appendix A (see supplementary material). We have not claimed any performance superiority between ResNEsts/A-ResNEsts and ResNets.
> >Moreover, methods/modifications that improve loss landscape of the SOTA models (e.g. ResNet) more often than not may not be useful at all, unless they could improve the original performance, or on par with it and improves the optimization, or simplifies the models and etc. As such, I do not support acceptance.
>
> As our title suggests, our intention is to propose analyzable block-based DNN models that exhibit provable theoretical guarantees, but not to improve any state-of-the-art models.
>
> Our ResNEst and A-ResNEst models are not meant to outperform ResNets or any SOTA models; instead, they are designed to circumvent mathematical intractabilities in ResNets so that we are able to derive theoretical guarantees for local minimizers. This is a significant first step to advance our understanding of why ResNets work so much better than just stacking layers in deep learning models. We hope our theoretical analysis will inspire more principle-guided approaches for block-based DNN model developments (e.g., ResNets, DenseNets, Transformers, U-Nets) in the future.
>
> The earliest theoretical work for ResNets can be dated back to [Hardt and Ma, 2016] which proved a vector-valued ResNet-like model using a linear residual function in each residual block has no spurious local minimum under squared loss and near-identity region assumptions. Since then, many influential ResNet theoretical works have been published based on different sets of assumptions [Li and Yuan, 2017, Liu et al., 2019, Li et al., 2018, Liang et al., 2018, Shamir, 2018, Kawaguchi and Bengio, 2019, Yun et al., 2019]. Our contributions advance the line of these works.
> >In what follows, I try my best to summarize the main contribution and ideas of the paper... could be replaced by a dense matrix $H_i$ ...
>
> The reviewer only tried to summarize Sections 2.2 and 2.3. These two subsections serve as background before main theorems. It is clear that the role of Proposition 1 is to help us understand the problem, but not serving as a property to be enjoyed (lines 176 to 196). The good properties that ResNEsts enjoy are Theorem 1, Remark 2, Corollary 1, and Theorem 2. The most important contributions in this paper are in Section 3 where we present the main theoretical results.
>
> >The authors proceed to conclude that the new optimization problem is convex, which seems obvious since the new model is a linear function function the hidden features. They then apply similar tricks to DenseNet.
>
> Our proposition 2 may look obvious, but it is a very important step towards deriving one of our main contributions, Theorem 1. The optimization problems ($P$) and ($P_{\\phi}$) are non-convex and high-dimensional, so results in Theorem 1 are nontrivial. To the best of our knowledge, it is the first theoretical guarantee for generic vector-valued L-block ResNet-like models (we call them ResNet-like because ResNEsts are different from ResNets, again the nonlinearity at the last representation can make a difference). The significance of Theorem 1 is stated in section 3 of our paper.
> >Theoretical understanding of ResNet is important and indeed very important! Principle-guided approach to design architecture is also equally important. However, I don't think the paper made significant advances in these two problems,
>
> The reviewer is absolutely correct that the theoretical understanding of ResNets is very important. To the best of our knowledge, our Theorem 1 establishes the first local minimizer guarantee that applies to generic vector-valued ResNet-like models that have an arbitrary number of residual blocks. Our contributions advance the line of ResNet theoretical works and generalize previous published theoretical results.
> >in particular, without strong empirical support. I would encourage the authors to provide experimental results to demonstrate the superior of the proposed architectures.
>
> We agree with the reviewer that empirical results for our new models would be interesting. We thank the reviewer for the encouragement. We evaluated ResNEsts, A-ResNEsts, and standard ResNets on CIFAR-10 and CIFAR-100 for 4 different architectures (different depth and width, they are WRN-40-4, WRN-16-8, ResNet-110, and ResNet-20). Please see our experimental results in the post above.
>
> ---
>
> Based on the significant contributions of the paper, we feel that the rating of 3 is too harsh and that the paper is deserving of a much higher rating. We sincerely hope the reviewer will reconsider based on the responses to the issues raised and the experimental evidence provided.

---

> > ### Comment · Reviewer_AWNq · 2021-08-19
> > **update**
> >
> > Thanks for the clarification and I appreciate the authors efforts in running extra experiments. In light of these, in particular, encouraging empirical results from A-ResNEsts, I increase my score to 5.
> >
> > I do appreciate the authors contribution in finding a mathematically more trackable model which also enjoys good empirical performance. The reason why I am still not convinced to accept the paper is
> >
> > (1)  The direction does not really touch generalization of ResNet, or explain why resnet **generalize** better than non-resnet family of architectures. In practice, both residual and non-residual networks can obtain good training accuracy/loss (e.g. 100% for cifar10), but the overall performance of residual network is better. Analyzing global minima of training loss won't lead us to the answers of this type of question.
> >
> > (2) I still find the paper hard to follow and improvement is needed. Note that there are many inline equations.

---

> > > ### Author Response · Authors · 2021-08-25
> > > **Clarification on the degradation problem and a new direction for better generalization**
> > >
> > > Thank you very much for your update. Our response is as follows:
> > >
> > > > Thanks for the clarification and I appreciate the authors efforts in running extra experiments. In light of these, in particular, encouraging empirical results from A-ResNEsts, I increase my score to 5.
> > >
> > > We thank the reviewer for the recognition of our experiments.
> > > >I do appreciate the authors contribution in finding a mathematically more trackable model which also enjoys good empirical performance. The reason why I am still not convinced to accept the paper is
> > > (1) The direction does not really touch generalization of ResNet, or explain why resnet generalize better than non-resnet family of architectures. In practice, both residual and non-residual networks can obtain good training accuracy/loss (e.g. 100% for cifar10), but the overall performance of residual network is better. Analyzing global minima of training loss won't lead us to the answers of this type of question.
> > >
> > > As we stated in lines 29 to 35 in the paper, stacking more and more layers in plain networks (non-residual networks) suffers from **worse training performance** (and thus worse testing performance). In contrast to plain networks, ResNets do not suffer from such a degradation problem. **The degradation problem is probably the most noticeable distinction between plain networks and ResNets.** To the best of our knowledge, how and why ResNets escape from the **degradation problem** (lines 31 to 35 in the paper) has not been well justified and still remains unknown or unclear. This paper theoretically answers these questions by proving the residual representation in the ResNEst model is guaranteed to be improved over blocks at any obtained local minimizer under practical conditions (Theorem 1 and Remark 2). Such a characterization on local minima is important and insightful because it guarantees stacking more and more blocks in ResNEsts always leads to better training performance.
> > >
> > > Our results are also consistent with advanced ResNet architecture designs. Our theory explains why **wide ResNets** are better than thin ResNets [Zagoruyko and Komodakis, 2016] (lines 120 to 123 and section 3 in the paper) and why **bottleneck blocks** are preferred over basic blocks (lines 269 to 275).
> > >
> > > Although the issue of generalization is not the focus of the paper, we agree with the reviewer that such a topic is also important. In fact, using these analyzable models and linear estimation viewpoints can also lead to new insights for better generalization in ResNets/ResNEsts/A-ResNEsts. As suggested in section 2.4 in the paper, linearly unpredictable nonlinear features $v_i$ in ResNEsts are necessary for strictly improved residual representations. Therefore, enforcing or **promoting the orthogonality** between all these nonlinear features can potentially be a new direction for improving the test performance or better generalization. For example, one can design an orthogonal regularizer to promote orthogonal nonlinear features in practice. Notice that such a new direction can be easily seen by our theory, and it is exclusive to ResNets/ResNEsts/A-ResNEsts (plain networks do not have a way to utilize the orthogonality).
> > > >(2) I still find the paper hard to follow and improvement is needed. Note that there are many inline equations.
> > >
> > > In addition to the improvements we mentioned in the first response to the reviewer, we will also move inline equations to their dedicated spaces as much as possible.

---

### Official Review · Reviewer_vBzc · 2021-07-27

**Rating:** 8
**Confidence:** 3

**Summary:**

This work studies the representational power of Residual Networks and try to explain theoretically a desirable property of using ResNets, namely: does a residual network guarantee that stacking more residual blocks does not lead to a degradation in performance of the network. The main approach of the authors is to view the output of a ResNet as an ensemble of several shallow networks as opposed to a non-ResNet which is simply a very deep composition of non-linear functions. Given this view point, the authors identify some simplifying assumptions which make the associated Empirical Risk Minimization problem tractable: 1) introducing a sequence of "de-coupling" weights (H) which allow the weights (W) in the ensemble to be independent of each other 2) removing the non-linearity at the end of the residual block and before the final affine transformation and 3) a few assumptions on the loss function (differentiability and convexity). With these assumptions, the authors prove a few desirable properties of ResNet architectures. Firstly, the authors show that stacking more residual blocks cannot hurt in the representational sense, and can improve if the new feature is linearly independent of the other features in the ensemble basis. Secondly, the authors show that "wide ResNets" always attain the empirical risk minimization lower bound, which has an implication on the quality of local optima attained by the ResNets. The authors also prove analogous results for a related class of architectures (DenseNets), in particular showing that DenseNets may be approximated by wide ResNests.

**Limitations And Societal Impact:**

Yes

**Main Review:**

This is a very interesting paper which studies an important problem from a theoretical angle, namely the representational power of Residual Networks, and what characterizes their empirical good performance compared to simply stacking dense functions. The work introduces some simplifications to make the problem more tractable, and show theoretically that in the setting when the final non-linearity is removed from a chain of residual blocks, we get a theoretical guarantee that stacking more residual blocks is guaranteed to not hurt performance in the optimization setting (modulo some assumptions). This seems like an important and very interesting observation, and I wonder if some of assumptions which the authors made to get to the theoretical result could lead us to design better real world architectures - for instance, would a "decoupled" ResNet lead to improvements over a regular ResNets empirically?

The paper is overall well written and easy to follow, although the main proofs of all the claims are in the Appendix which doesn't seem included. However, with the assumptions introduced by the authors, the conclusions of the theorems sound reasonable. In terms of future directions/improvements, I would encourage the authors to investigate this phenomenon empirically as well, to see if some of the assumptions hold in real world datasets, which would make the work more compelling.

Typos/Clarifications:

Line 234: It seems N_0 has not been defined before, is it some large constant we are assured the existence of?

**Time Spent Reviewing:**

4

---

> ### Author Response · Authors · 2021-08-10
> **Experimental results and clarification on proofs and definitions**
>
> Thank you very much for your comments that will most certainly help us improve the paper, given the opportunity. We will make necessary modifications to the main text and add empirical results in the supplementary material as we revise the manuscript.
>
> ### Below we respond to the reviewer's specific suggestions and concerns:
> >This seems like an important and very interesting observation, and I wonder if some of assumptions which the authors made to get to the theoretical result could lead us to design better real world architectures - for instance, would a "decoupled" ResNet lead to improvements over a regular ResNets empirically?
>
> We thank the reviewer for these important suggestions. We strongly agree with the reviewer that empirical results for our new models would be helpful to support the derived theoretical guarantees.
>
> We evaluated ResNEsts, A-ResNEsts, and standard ResNets on CIFAR-10 and CIFAR-100 for 4 different architectures (different depth and width, they are WRN-40-4, WRN-16-8, ResNet-110, and ResNet-20). Please see our experimental results in the post above.
>
> >The paper is overall well written and easy to follow, although the main proofs of all the claims are in the Appendix which doesn't seem included.
>
> The proofs for all theorems, corollaries, and propositions in the paper were already included in the Supplementary Material when we submitted the paper (we called it Appendix). We are sorry about such confusion. We will refer to Supplementary Material instead of Appendix in the paper as we revise the manuscript.
> >In terms of future directions/improvements, I would encourage the authors to investigate this phenomenon empirically as well, to see if some of the assumptions hold in real world datasets, which would make the work more compelling.
>
> We thank the reviewer for the encouragement on our future directions. Assumption 2 and Assumption 3 are practical because they are not limited to specific loss functions, residual functions, or datasets. Their justifications are in lines 218 to 220, and lines 261 to 264, respectively. The most interesting assumption is probably Assumption 4 as it requires the bottleneck condition (an implicit condition behind Assumption 4, see lines 252 to 260) to be satisfied. The justification of Assumption 4 is given in lines 265 to 267. One of our future directions is to relax Assumption 4 so that our theoretical results can be also applied to basic blocks (see Figure 1 in the wide ResNet paper by Zagoruyko and Komodakis). However, such a relaxation may require imposing additional assumptions on residual functions or datasets. Another future direction is to generalize our local minimum guarantees to other architectures that also utilize skip connections. For example, Transformers, U-Nets/V-Nets, etc.
> >Line 234: It seems N_0 has not been defined before, is it some large constant we are assured the existence of?
>
> $N_O$ is defined in line 132. It is the output dimension of the network which is equivalent to the number of rows in $W_{L+1}$. We will point out where the definition is in Assumption 3 as we revise the manuscript.

---

> ### Author Response · Authors · 2021-08-30
> **We would be happy to answer any further questions**
>
> Once again, thank you for your positive review. It was encouraging. We have provided experimental results and they are quite promising. We hope you find them also interesting; and, reinforce your positive view of the paper and make you more supportive of our work.

---

### Author Response · Authors · 2021-08-12
**Experimental results**

## Clarification
1. Although this is a theoretical paper, we strongly agree with the reviewers that empirical results are helpful to understand our new models, ResNEsts and A-ResNEsts. How do these models perform compared to standard ResNets?
2. How do standard ResNets perform when the ReLU in the final residual representation is dropped?
3. Our ResNEst and A-ResNEst models are not meant to outperform ResNets or any SOTA models; instead, they are designed to circumvent mathematical intractabilities in ResNets so that we are able to derive theoretical guarantees for local minimizers.
4. We will add empirical results in the supplementary material as we revise the manuscript.

## Architectures
Every ResNEst was a standard ResNet without the batch normalization and ReLU at the final residual representation. Every BN-ResNEst was a standard ResNet without the ReLU at the final residual representation. In other words, a BN-ResNEst is a modified ResNEst because it adds a batch normalization layer at the final residual representation in the ResNEst. Such a modification can avoid gradient explosion during training and allow larger learning rates to be used.
For A-ResNEsts, we applied 2-dimensional average pooling on each $\\mathbf{v}_i$ going into each $H_i$. The standard ResNets used are wide ResNet 16-8 (WRN-16-8), WRN-40-4 [Zagoruyko and Komodakis, 2016], ResNet-110, and ResNet-20 [He et al., 2016].

## Implementation details
The training procedure is exactly the same as the wide ResNet paper by Zagoruyko and Komodakis.
The loss function was a cross-entropy loss. The batchsize was $128$.
The learning rate was initially set to $0.1$ ($0.01$ for ResNEsts to avoid gradient explosion [^footnote1]) and decreased by a factor of $5$ after training $60$, $120$, and $160$ epochs. All networks were trained for $200$ epochs in total. The optimizer was stochastic gradient descent (SGD) with Nesterov momentum. The momentum was set to $0.9$. The weight decay was $0.0005$. Our code will be available in the supplementary material.

## Tables of test accuracy on CIFAR-10 and CIFAR-100

Our empirical results are shown in the two tables below (classification accuracy and number of parameters). The classification accuracy is an average of $7$ trials with different initializations.

### CIFAR-10

| Architecture | Standard | ResNEst | BN-ResNEst | A-ResNEst |
| :-: | :-: | :-: | :-: | :-: |
|WRN-16-8| $95.68\\%$ (11M) | $94.59\\%$ (11M) | $95.59\\%$ (11M) | $95.39\\%$ (8.7M) |
|WRN-40-4| $95.61\\%$ (9.0M) | $94.74\\%$ (9.0M) | $95.72\\%$ (9.0M) | $95.59\\%$ (8.4M) |
|ResNet-110| $94.47\\%$ (1.7M) | $92.78\\%$ (1.7M) | $94.63\\%$ (1.7M) | $94.22\\%$ (1.7M)|
|ResNet-20| $92.83\\%$ (0.27M) | $91.20\\%$ (0.27M) | $92.80\\%$ (0.27M) | $92.68\\%$ (0.24M) |

### CIFAR-100

| Architecture | Standard | ResNEst | BN-ResNEst | A-ResNEst |
| :-: | :-: | :-: | :-: | :-: |
|WRN-16-8| $79.44\\%$ (11M) | $75.61\\%$ (11M) | $79.26\\%$ (11M)| $79.01\\%$ (8.9M) |
|WRN-40-4| $79.04\\%$ (9.0M) | $75.49\\%$ (9.0M) | $79.21\\%$ (9.0M) | $78.64\\%$ (8.7M) |
|ResNet-110| $74.20\\%$ (1.7M) | $69.65\\%$ (1.7M) | $74.50\\%$ (1.7M) | $73.21\\%$ (1.9M)|
|ResNet-20| $69.51\\%$ (0.28M) | $65.19\\%$ (0.28M) | $69.25\\%$ (0.28M) | $68.64\\%$ (0.27M)|

## A-ResNEsts empirically exhibit competitive performance to standard ResNets
In general, A-ResNEsts exhibit competitive classification accuracy with fewer parameters compared to standard ResNets; and ResNEsts are not as good as A-ResNEsts. The A-ResNets have fewer parameters than the ResNEsts and standard ResNets (^footnote2) because A-ResNEsts do not have the layers $W_L$ and $W_{L+1}$; and the number of prediction weights is usually not larger than the number of weights in $W_L$ and $W_{L+1}$ (see Figure 1 and compare it with Figure 2).

## A BN-ResNEst slightly outperforms a standard ResNet when the network is very deep
BN-ResNEsts slightly outperform standard ResNets and A-ResNEsts in architectures WRN-40-4 and ResNet-110 on both CIFAR-10 and CIFAR-100. For architectures WRN-16-8 and ResNet-20, BN-ResNEsts remain competitive performance to standard ResNets. According to the above experimental results, **simply dropping the ReLU at the final residual representation can even be beneficial for very deep ResNets**.

---

[^footnote1]: Learning rates $0.1$ and $0.05$ both led to gradient explosion in training ResNEsts, so we used $0.01$ for ResNEsts to avoid divergence. A-ResNEsts and BN-ResNEsts do not have such an issue.

[^footnote2]: A-ResNEsts can have more parameters than standard ResNets when the depth and the output dimension are very large. For example, the A-ResNEst using the ResNet-110 architecture in CIFAR-100.

---

> ### Comment · Reviewer_AWNq · 2021-08-19
> **Results on ImageNet**
>
> Just want to provide an extra data point on ImageNet. The experiments uses the code: https://github.com/google/flax/blob/main/examples/imagenet/models.py
>
> Original accuracy on ResNet-50 (3 runs): **0.766, 0.766, 0.768**
>
> With modification:  **0.744, 0.745, 0.745**
>
> Here is what the modification means: replacing the `self.act` by the `identity` function in  https://github.com/google/flax/blob/main/examples/imagenet/models.py#L79. I think the modification is close to what ResNEst in the paper. Note that there is **always** an activation after the residual connection in the above ResNet code.
>
> We see there is a 2.3% drop when removing the `self.act` which is aligned with the above results in CIFAR-10/100, in which ResNEst perform noticeably worse.  Unfortunately, I don't have data point for the A-ResNEst.

---

> > ### Author Response · Authors · 2021-08-25
> > **Clarification on the definition of ResNEsts and BN-ResNEsts**
> >
> > We thank the reviewer for providing the above ImageNet results on the ResNet-50 and modified ResNet-50. After checking the code, we found that their models are very different from the models we consider in the paper and experiments. Below we address the differences.
> >
> > ## Our models use pre-activation residual blocks
> > According to the code in the link above, these ResNets are in the original (non-pre-activation) form [^1] (Figure 4(a) in [^2]), not in the pre-activation form [^2] (Figure 4(e) in [^2]). In [^2], He et al. showed that pre-activation ResNets are superior to ResNets in the original form. Our models in the paper and experiments all use pre-activation residual blocks (line 62, Figure 2, and lines 124 to 133), i.e., they are in the pre-activation form.
> >
> > ## BN-ResNEsts drop the ReLU and ResNEsts drop both the BN and ReLU
> > The definition of ResNEsts and BN-ResNEsts are given in the above post “Experimental results.” The experimental results show that keeping the batch normalization and simply dropping the ReLU at the final residual representation in standard pre-activation ResNets can improve the test accuracy in CIFAR-10 and CIFAR-100 when the depth is large. In other words, BN-ResNEsts outperform standard ResNets when the network is very deep. However, if the batch normalization at the final residual representation is also dropped, the test accuracy is noticeably lower.
> >
> > ---
> >
> > [^1]: K. He, X. Zhang, S. Ren, and J. Sun. Deep residual learning for image recognition. In Conference on Computer Vision and Pattern Recognition, pages 770–778. IEEE, 2016a
> >
> > [^2]: K. He, X. Zhang, S. Ren, and J. Sun. Identity mappings in deep residual networks. In European Conference on Computer Vision, pages 630–645. Springer, 2016b.

---

### Decision · Program_Chairs · 2021-09-27

**Decision:**

Accept (Poster)

**Comment:**

This work studies the representational power of ResNets and aims to answer the question of when adding more residual blocks in a ResNet can be guaranteed to not lead to a degradation in performance. All 5 reviewers find the paper interesting and think its contribution is a valuable addition to the literature. All reviewers were active in discussion with the authors and with the committee. No serious concerns surfaced that would invalidate the contribution made by the paper. After committee discussion the reviewers reached a consensus recommendation to accept the paper.

Multiple reviewers indicated that the author response was especially useful and served to clear up some minor concerns that existed after the first round of reviews. Authors, please integrate this discussion into the camera ready version of the paper.